# Nucleotide binding by the widespread high-affinity cyclic di-GMP receptor MshEN domain

Yu-Chuan Wang[1,*], Ko-Hsin Chin[2,*], Zhi-Le Tu[1], Jin He[3], Christopher J. Jones[4], David Zamorano Sanchez[4], Fitnat H. Yildiz[4], Michael Y. Galperin[5] & Shan-Ho Chou[1,2]

C-di-GMP is a bacterial second messenger regulating various cellular functions. Many bacteria contain c-di-GMP-metabolizing enzymes but lack known c-di-GMP receptors. Recently, two MshE-type ATPases associated with bacterial type II secretion system and type IV pilus formation were shown to specifically bind c-di-GMP. Here we report crystal structure of the MshE N-terminal domain (MshEN$_{1-145}$) from *Vibrio cholerae* in complex with c-di-GMP at a 1.37 Å resolution. This structure reveals a unique c-di-GMP-binding mode, featuring a tandem array of two highly conserved binding motifs, each comprising a 24-residue sequence RLGxx(*L/V/I*)(*L/V/I*)xxG(*L/V/I*)(*L/V/I*)xxxxLxxxLxxQ that binds half of the c-di-GMP molecule, primarily through hydrophobic interactions. Mutating these highly conserved residues markedly reduces c-di-GMP binding and biofilm formation by *V. cholerae*. This c-di-GMP-binding motif is present in diverse bacterial proteins exhibiting binding affinities ranging from 0.5 µM to as low as 14 nM. The MshEN domain contains the longest nucleotide-binding motif reported to date.

[1] Institute of Biochemistry, National Chung Hsing University, Taichung 40227, Taiwan, Republic of China. [2] Agricultural Biotechnology Center, National Chung Hsing University, Taichung 40227, Taiwan, Republic of China. [3] State Key Laboratory of Agricultural Microbiology, College of Life Science and Technology, Huazhong Agricultural University, Wuhan, Hubei 430070, People's Republic of China. [4] Department of Microbiology and Environmental Toxicology, University of California, Santa Cruz, Santa Cruz, California 95064, USA. [5] National Center for Biotechnology Information, National Library of Medicine, National Institutes of Health, Bethesda, Maryland 20894, USA. * These authors contributed equally to this work. Correspondence and requests for materials should be addressed to S.-H.C. (email: shchou@nchu.edu.tw).

Cyclic dimeric GMP (c-di-GMP) is a second messenger controlling many important cellular activities, such as biofilm formation, biogenesis and action of flagella and pili, and secretion of pathogenic factors in diverse bacteria[1]. While the mechanisms of c-di-GMP biosynthesis and degradation have been characterized in some detail, the nature of c-di-GMP receptors and the mechanisms of c-di-GMP-mediated regulation are still not fully understood. In the past several years, a variety of c-di-GMP-binding protein receptors[2] and several classes of riboswitches[3] have been described. Yet since the number of c-di-GMP-metabolizing enzymes (diguanylate cyclases and phosphodiesterases) encoded in any given genome is far larger than that of known c-di-GMP receptors, it is likely that more receptors remain to be identified.

Recently, MshE (locus tag VC0405), an ATPase associated with the mannose-sensitive haemagglutinin type IV pilus (T4P) formation in *Vibrio cholerae*, as well as its homologue PA14_29490, an ATPase involved in the bacterial type II secretion system (T2SS) in *Pseudomonas aeruginosa*, were shown to be potent c-di-GMP-binding proteins[4,5]. In both proteins, c-di-GMP binding has been localized to their N-terminal T2SSE_N (hereafter, MshEN) domains, whereas their C-terminal ATPase domains were shown to bind ATP but not c-di-GMP[5]. Two residues, Arg9 and Gln32 of MshE, were identified as involved in c-di-GMP binding[5] but these residues did not belong to any of the previously described canonical c-di-GMP binding motif[2]. To explore the possibility that MshE contains a special kind of c-di-GMP-binding motif, we have solved the crystal structure of its MshEN domain (*Vc*MshEN) in complex with c-di-GMP to a nearly-atomic level resolution of 1.37 Å. The structure revealed two 24-residue motifs linked together by 5 non-conserved residues that work cooperatively to form a 53-residue domain that binds a complete c-di-GMP molecule. This c-di-GMP binding domain is found in a number of bacteria lacking any c-di-GMP receptors discovered to date, and can be fused with ATPase, glycosyltransferase and other domains. Several such sequences have been tested and found to bind c-di-GMP with high affinity (with $K_D$ ranging from 14 nM to 0.5 μM). These results establish MshEN as a widespread and highly sensitive c-di-GMP-binding receptor domain, and open new avenues towards uncovering additional c-di-GMP-mediated functions in bacteria.

## Results

**Structural characterization of the MshEN-c-di-GMP complex.** The *Vc*MshEN structure was well-refined, and its macromolecular chain could be traced from residue Lys5 to Tyr145 without any interruption. Ramachandran plot showed that all torsional angles of MshEN-c-di-GMP complex were within the acceptable regions without any outliers (detailed data are in Table 1). Each MshEN structure could be further divided into two subdomains, a four-helix MshEN_N ($\alpha 1 - \alpha 4$) and an $\alpha/\beta$ MshEN_C (a $\beta 1 - \beta 3$ antiparallel $\beta$-sheet surrounded by three helices) (Fig. 1a). C-di-GMP was bound mainly in the MshEN_N sub-domain, with the only contribution from the MshEN_C subdomain coming from Asp108, which interacts with the amino group of the second guanine (Gua2) base of c-di-GMP (Figs 1a and 2c).

The *Vc*MshEN_N subdomain (MshE$_{1-64}$) comprises a four-helix bundle, which is similar to the previously described N-terminal part of the XpsEN domain[6] from *Xanthomonas campestris* strain 17 (ref. 7). However, superposition of the *Vc*MshEN and *Xc*XpsEN structures revealed certain differences (with a r.m.s.d. of 1.35 Å over 56 residues, Fig. 1b). In *Vc*MshEN, the N-terminal $\alpha 1$ helix is broken since the full $\alpha 1$ helix causes significant steric clash against c-di-GMP (cyan arrow).

In addition, $\alpha 2$ helix in *Vc*MshEN is extended by two turns and swings downward to enhance interaction with c-di-GMP (blue arrow). The ligand c-di-GMP was clearly visible in the *Vc*MshEN domain, as shown in its non-biased Fo-Fc electron density map at 3σ (Fig. 1c). Unlike the stacked conformation usually adopted for binding with PilZ domain receptor proteins (2RDE in Fig. 1d), or the extended form in the EAL domain proteins (3PJT), c-di-GMP in the *Vc*MshEN complex adopts a bulged conformation more akin to the one in the *Xc*FimX$^{EAL}$-PilZ complex (4F3H, drawn in yellow carbons in Fig. 1d)[8]. When plotted in electrostatic surface charge format, it is clear that the MshEN_N subdomain contains a positively charged patch comprising the Lys5, Arg7, Lys8, Arg9 and Arg38 residues (Fig. 1e). Interaction of the negatively charged c-di-GMP with the positively charged N-terminal region of MshEN could play a role in stabilizing the complex. The first guanine base (Gua1) of c-di-GMP is stacked by the side chain atoms of Arg9 on one side, and by the hydrophobic residues Leu25, Leu29 and Leu39 on the other side, being sandwiched between the two hydrophobic surfaces (Fig. 1c). The second guanine base (Gua2) of c-di-GMP is bound in a similar way as described below.

**MshEN binds c-di-GMP via a 53-residue-long domain.** Although c-di-GMP conformation in the MshEN complex is similar to that in the EAL domain in the *Xc*FimX$^{EAL}$-c-di-GMP complex (PDB: 4F3H)[8], MshEN contains a ligand-binding sequence that is dramatically different from the ExLxR motif in the EAL domain[2]. In *Vc*MshEN, c-di-GMP is accommodated by two 24-residue motifs [**RLG**xx(*L*)(*V/I*)xx**G**(*I/F*)(*L/V*)xxxx**L**xxx **L**xx**Q**] that are linked by a 5-residue spacer to form a complete 53-residue-long domain (Fig. 2a). The six residues marked in bold in each motif are highly conserved and play key roles in c-di-GMP binding. For example, the first Arg residue in the $^9$RLG$^{11}$ sequence (conserved residues shown in red in Fig. 2a,c) interacts with the guanine base Gua1 of c-di-GMP, albeit not via Hoogsteen-edge binding[2], but via stacking of its side chain atoms along the Gua1 base. The second Leu residue of the motif (Leu10) forms a triangular hydrophobic cluster with two other Leu residues (Leu54 and Leu58) located in the **L**xxx**L**xx**Q** sequence of the second motif (conserved residues shown in blue in Fig. 2a); this cluster interacts with Gua2 base through extensive hydrophobic CH-π interactions (Fig. 3). The importance of CH-π interaction in stabilizing protein or protein–ligand structure has been noted before[9]. The backbone amide protons of the third Gly residue and the following non-conserved residue in the motif form two hydrogen bonds (H-bonds) with the O$^6$ and N$^7$ atoms of Gua1 via its Hoogsteen edge (Fig. 3a). The central hydrophobic residues (shown above in italics) do not directly interact with the guanine base, but form a hydrophobic core to stabilize the MshEN_N four-helix bundle structure (Fig. 2c). The Gly residues (G18 and G47) in the 53-residue domain are also highly conserved and are required for the turns between the $\alpha 1 - \alpha 2$ and $\alpha 3 - \alpha 4$ helices (Fig. 2b). Finally, the two Leu residues shown in bold in the last section of the **L**xxx**L**xx**Q** sequence in the second 24-residue motif (Leu54 and Leu58) participate in forming a triangular hydrophobic cluster with Leu10 from the first motif as described above (see Fig. 3b) to interact with the Gua2 base. Each of the two conserved Gln residues (Gln32 and Gln61) in the motif forms two H-bonds with the ribose 3'-OH and phosphate oxygen atom (Fig. 2c). Only these Gln residues and the two helical residues (the third Gly and the subsequent non-conserved residue) interact with c-di-GMP via polar interactions. All other conserved residues participate in forming extensive hydrophobic interactions with the guanine base, an arrangement that to our knowledge has not been seen before.

**Table 1 | Statistics of data collection and refinement statistics for SAD (SeMet) structures.**

| | VcMshEN-c-di-GMP | Se-Met-VcMshEN-c-di-GMP |
|---|---|---|
| Data collection | | |
| Space group | $P2_12_12_1$ | $P2_12_12_1$ |
| Cell dimensions | | |
| a, b, c (Å) | a = 52.852, b = 61.862, c = 75.621 | a = 52.815, b = 59.583, c = 81.66 |
| α, β, γ (°) | α = β = γ = 90° | α = β = γ = 90° |
| Resolution (Å) | 50 − 1.37 (1.42 − 1.37)* | 50 − 1.83 (1.9 − 1.83)* |
| $R_{merge}$ | 3.8 (31.4)* | 3.5 (26.7)* |
| $I/\sigma I$ | 18.2 (2.4)* | 20.6 (3.7)* |
| Completeness (%) | 95.4 (99.7)* | 91.2 (97.6)* |
| Redundancy | 4.6 (4.8)* | 8.6 (9)* |
| | | |
| Refinement | | |
| Resolution (Å) | 50 − 1.37 (1.42 − 1.37)* | |
| No. reflections | 234156 (50359)* | |
| $R_{work}/R_{free}$ | 20.2/22.5 | |
| No. residues or atoms | | |
| Protein | 278 | |
| Ligand (c-di-GMP) | 2 | |
| Water | 362 | |
| B-factors | | |
| Protein | 20.1 | |
| Ligand (c-di-GMP) | 14.9 | |
| Water | 26.3 | |
| R.m.s deviations | | |
| Bond lengths (Å) | 0.0065 | |
| Bond angles (°) | 1.286 | |

**Identical binding modes of the two guanine bases of c-di-GMP.** Since a rather long motif is involved in binding a single c-di-GMP molecule, it is intriguing to compare how the two 24-residue motifs bind each respective guanine base. We thus superimposed the Gua1 binding residues (from Arg9 to Gln32) with the Gua2-binding residues (from Arg38 to Gln 61) as shown in Supplementary Fig. 5. From the figure, one can clearly see that, although the two guanine bases are located in a different environment (Fig. 2a,c), the Cα atoms of the conserved binding residues are well-aligned, with a r.m.s.d. of only 0.8 Å over all 25 residues. The entire α1-linker-α2 segment superimposes well with the α3-linker-α4 segment, with most of the side chain atoms of the conserved residues including those from Leu10/Gly11/Asp12/Leu25/Gln32 also aligning with those from Leu39/Gly40/Asp41/Leu54/Gln61. Only the side chains of the Arg9/Arg38 and Leu29/Leu58 residues exhibit some differences.

**The MshEN domain is specific for c-di-GMP.** From the crystal structure, it is clear that there is a close fit between the two backbone amide protons (H-bond donors) of Gly11/Gly40 and Asp12/Asp42 with the guanine base $O^6$ and $N^7$ atoms (H-bond acceptors) (Fig. 3a). When c-di-GMP is changed to c-di-AMP, the corresponding H-bond acceptor $O^6$ atom (marked by a magenta arrow) will be converted into a bulky amino group (an H-bond donor), breaking the complementarity and causing a considerable clash with the Asp12 amide proton. Two such serious $O^6 − NH_2^2$ steric clashes in each motif are probably the reason for the lack of any c-di-AMP binding by the MshEN domain, as checked by isothermal calorimetry (ITC) (Fig. 1g). The excellent fit of c-di-GMP within the motif also explains why the MshEN domain cannot bind cGMP, GMP, GDP or GTP, or the adenine base-containing nucleotides such as cAMP, ATP, ADP and AMP, as has been reported previously[5] and also confirmed here using differential scanning fluorimetry (DSF, Supplementary Fig. 2).

**MshEN can also bind cGAMP.** In contrast to c-di-AMP, the (3′–5′)-cyclic AMP-GMP (cGAMP) molecule is also present in V. cholerae[10], and was found to bind the MshEN_N domain with a reasonable affinity ($K_D$ $3.3 \times 10^{-4}$ M, Fig. 1h). This binding constant is between those observed with c-di-GMP ($K_D$ $5.0 \times 10^{-7}$ M, Fig. 1f) and c-di-AMP (non-detectable). Thus, the stability and affinity of the MshEN_N-cGAMP complex, although not as strong as those of the MshEN_N-c-di-GMP complex, are still discernable and cannot be ignored. It remains to be seen whether cGAMP interacts with MshEN strongly enough to exhibit a specific function in vivo (for example, when V. cholerae enters the host intestinal environment[10]).

**Conserved residues are required for c-di-GMP binding.** Since the high-resolution complex structure showed that many residues are involved in binding c-di-GMP, we have prepared a series of single, double or triple MshEN_N variants to check their contribution towards c-di-GMP binding using ITC. Biophysical analysis confirmed that hydrophobic interactions are indeed important in binding c-di-GMP (Supplementary Table 1), since mutating any one of these conserved hydrophobic residues resulted in a larger than 10-fold decrease in binding affinity. Alteration of some charged residues in MshEN, including Arg9, Asp12 and Gln32, and their effects on the c-di-GMP binding have been reported in a previous study[5]; however, the binding affinity did not seem to change more than fivefold[5].

In addition to the affinity measurements, we have also checked the stabilities of native MshEN_N and its representative active site variants by using the DSF method in the absence or presence of c-di-GMP (Supplementary Table 2). The wild-type and all tested representative variants had their Tm values in the 46–60 °C range, indicating that these variants are stable in the absence of c-di-GMP. For the G40I and G11L variants, the presence of c-di-GMP did not change their Tm values, consistent with the poor, if any, binding of the ligand (see Supplementary Table 2).

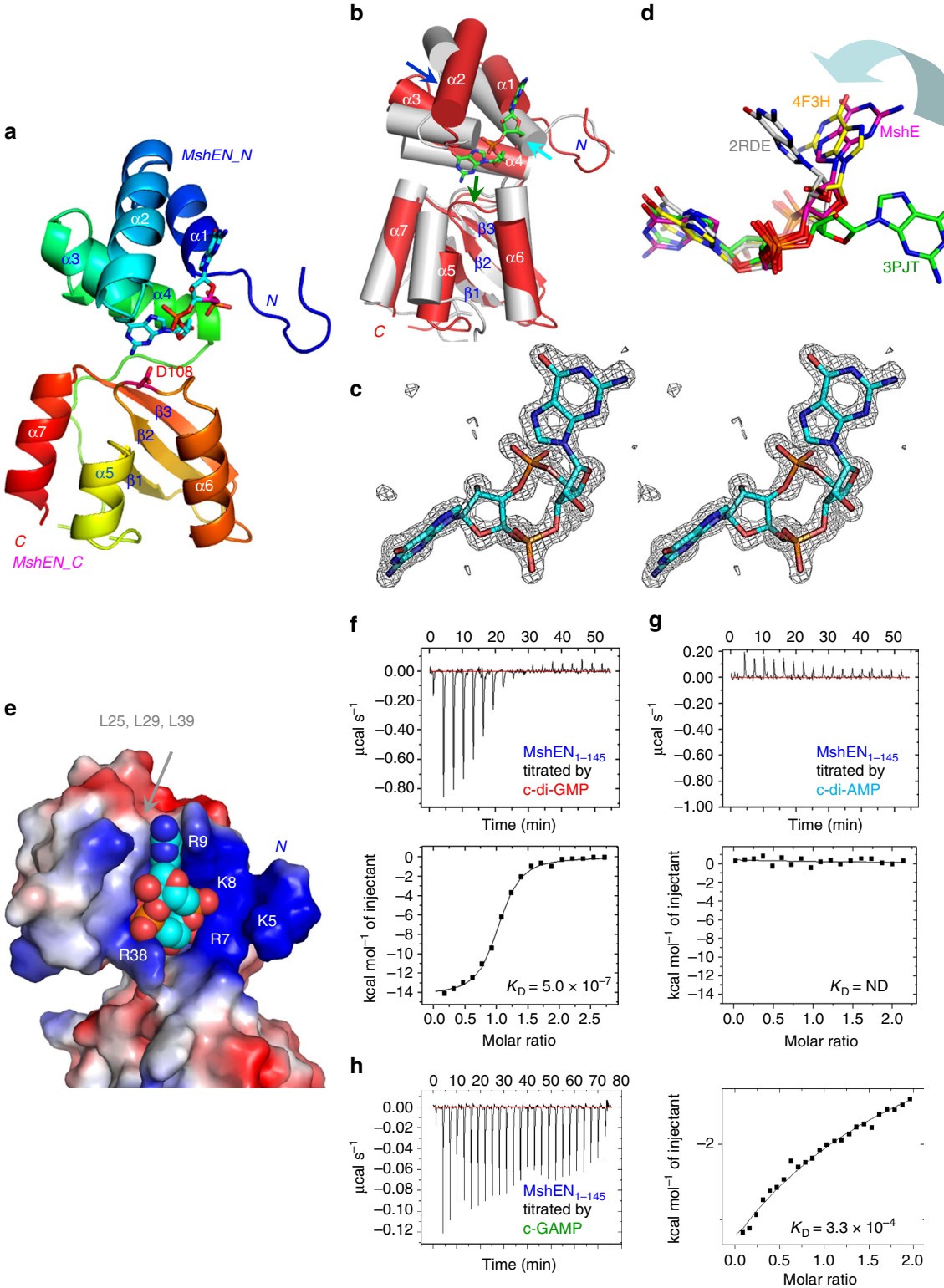

**Figure 1 | Crystal structure of the MshEN-c-di-GMP complex.** (**a**) A bi-domain structure, with c-di-GMP (drawn in stick representation with carbons in cyan) bound mainly by the MshEN_N domain. (**b**) Superposition of MshEN-c-di-GMP (in red, c-di-GMP carbons are in green) and XpsEN (in grey) structures. Structural adjustments required for c-di-GMP binding are marked by cyan, blue and green arrows. (**c**) Well-defined non-biased Fo-Fc electron density map of the c-di-GMP molecule drawn at $3.0\sigma$ level *in stereo*. (**d**) C-di-GMP in the MshEN-c-di-GMP complex adopts a bulged conformation similar to that in the FimX$^{EAL}$-PilZ complex (4F3H). A representative stacked form (2RDE) and a fully extended form (3PJT) are shown for comparison. (**e**) MshEN_N domain exhibits a positively charged patch comprising several N-terminal Arg and Lys residues (indicated by white letters) to facilitate c-di-GMP binding. Gua1 base stacks with Arg9 side chain atoms in one side and the Leu25, Leu29 and Leu39 hydrophobic cluster in the other side. (**f**) Tight binding between *Vc*MshEN and c-di-GMP as reflected by ITC measurement. (**g**) *Vc*MshEN is specific for c-di-GMP; no detectable binding of c-di-AMP was observed by ITC. (**h**) *Vc*MshEN exhibited a weaker binding with cGAMP was observed by ITC.

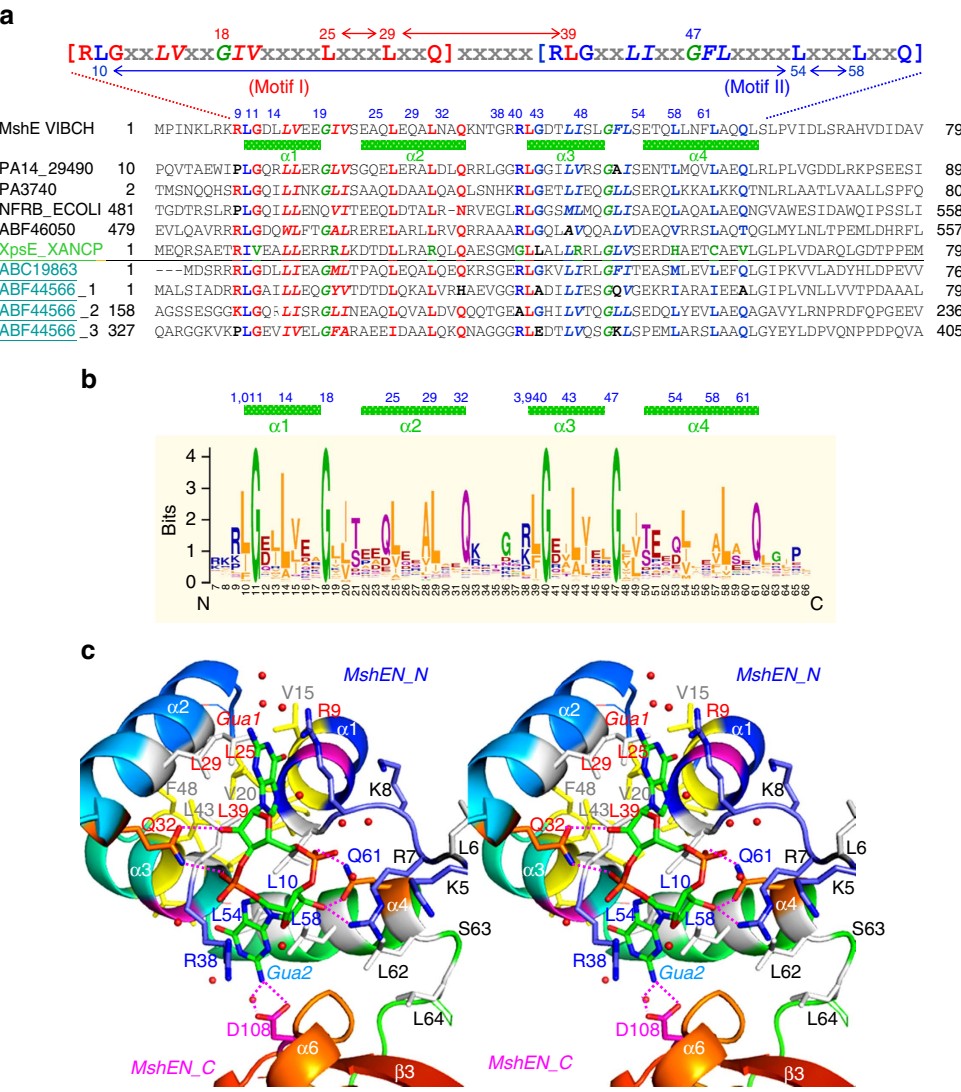

**Figure 2 | Mechanism of c-di-GMP binding by the MshEN domain.** (**a**) Sequence alignment of the c-di-GMP-binding motifs. The top line shows the unique 53-residue-long sequence comprising 2 similar motifs (conserved residues in motif I are shown in red and those in motif II are in blue) separated by a 5-residue linker. Leu10 belongs to motif II and Leu39 belongs to motif I. The three Leu residues forming the hydrophobic cluster of motif I are connected by red double-edge arrows, those in the motif II by blue double-edge arrows. GspE from Xcc17 (PDB: 2D27, in green) has lost the ability to bind c-di-GMP due to the changes in several key conserved residues. Other proteins are listed under their GenBank or UniProt identifiers. (**b**) Sequence logo generated from an alignment of 2,021 non-redundant MshEN sequences that contain glycines in positions 11, 18, 40 and 47 (VcMshEN numbering). (**c**) Stereo picture of the MshEN-c-di-GMP complex. Water oxygen atoms are shown as red balls. Gua1 base is on the top and interacts with residues in motif I (residues labelled in red), while Gua2 base is at the bottom and interacts with residues in motif II (residues labelled in blue). Asp108 (in pink) from the MshEN_C domain is also involved in binding c-di-GMP.

For the single L10A, L25A, or other multiple L54A/L58A or L10A/L54A/L58A variants that preserve the hydrophobic character of the active site residues, the Tm values increased by $\geqslant 5\,^\circ\text{C}$ in the presence of c-di-GMP, meaning that these variants could still be stabilized by c-di-GMP binding.

Taken together, the above described unique interaction mode leads to a strong binding between VcMshEN and c-di-GMP. The dissociation constant $K_D$ of 0.5 μM for VsMshEN (Fig. 1f) is at the lower end of c-di-GMP concentrations present in the host cells (0.05 to 10 μM)[1,11]. This unique c-di-GMP-binding mode is also generally 10-fold stronger than those of the PilZ domains. In fact, the binding affinity of the MshEN_N domain can even reach to $K_D$ of 14 nM when fused with a different MshEN_C domain (Supplementary Table 6; Supplementary Fig. 4).

**Biofilm formation by *V. cholerae* MshE variants**. We next evaluated the biological significance of the amino acids that engage in hydrophobic interactions with c-di-GMP within the VcMshEN by analysing different phenotypes associated with MshE function. We first established that the wild-type *V. cholerae* and mutant strains exhibit similar growth rates before the biofilm experiments (Supplementary Fig. 7). As previously reported, strains lacking *mshE* are defective in surface pilin production, initial stages of surface attachment and hence formation of mature biofilm; they also have increased motility in soft agar plates compared with the parental strain[4]. These phenotypes could be complemented by the expression of *mshE in trans*. Single substitutions in Leu10 (L10A), Gly11 (G11I) and Leu58 (L58A) affect the ability of respective MshE variants to complement the *mshE* mutant (Fig. 4). The mutation in G11

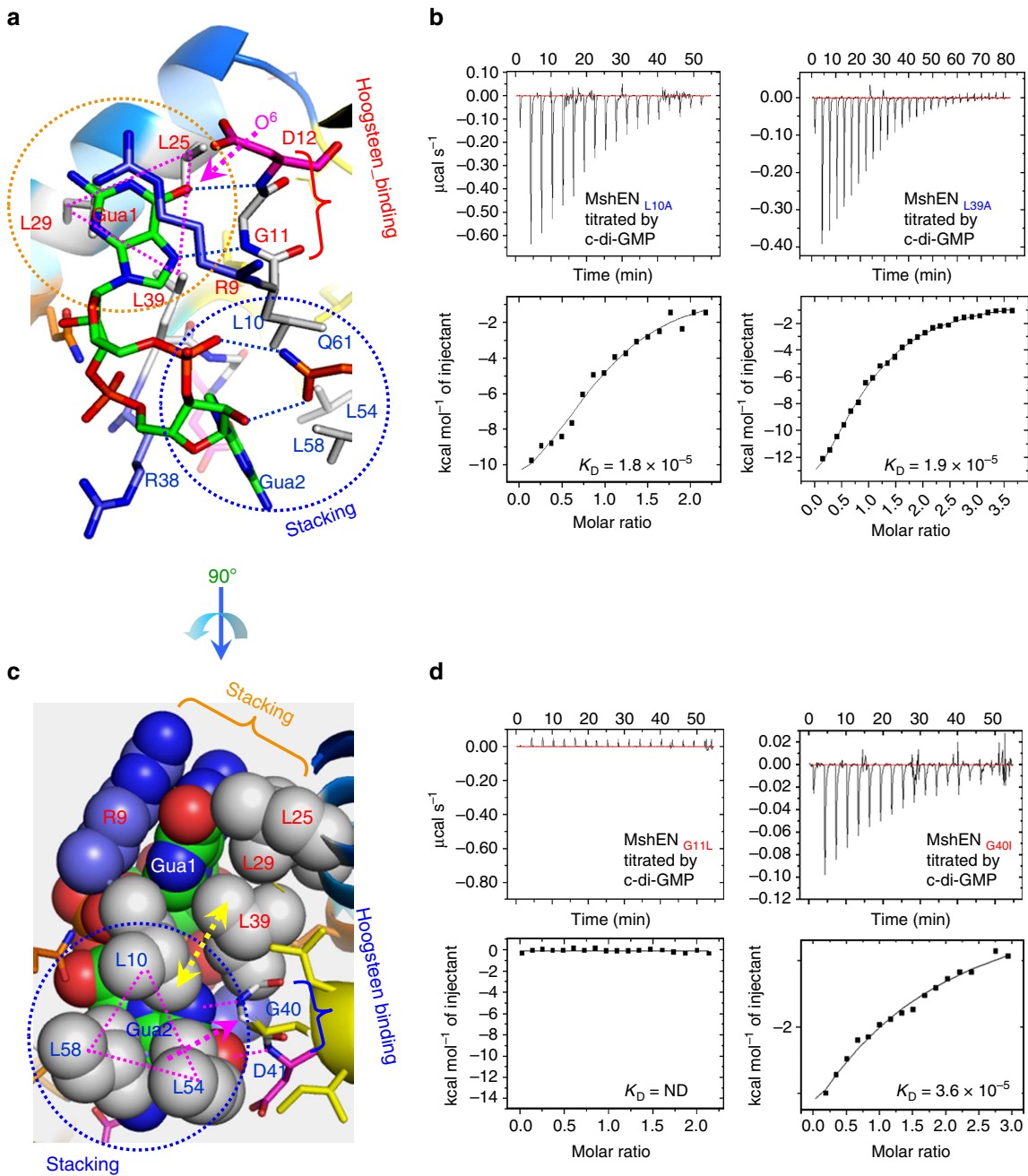

**Figure 3 | Roles of hydrophobic interactions in c-di-GMP binding. (a)** Positions of the residues in the [9]RLGD[12] sequence relative to the c-di-GMP molecule. Specific binding of c-di-GMP is due to the pairing of its O[6] atom (indicated by a magenta arrow) with the Asp12 amide proton. Replacement of the O[6] atom with $NH_2^2$ of c-di-AMP would cause a significant steric clash. **(b)** Single amino acid change of Leu10 or Leu39 to Ala causes considerable reduction of c-di-GMP-binding affinity as measured by ITC. **(c)** A unique hydrophobic triangular stacking of three conserved Leu residues with the guanine base (drawn as spheres). Leu10-Leu54-Leu58 residues form a triangular stack with Gua2 base. Similarly, Leu25-Leu29-Leu39 residues form another triangular stack with Gua1 base. **(d)** Single amino acid change of the conserved middle Gly residue in the loops connecting helices $\alpha1 - \alpha2$ and $\alpha3 - \alpha4$ to Leu or Ile also causes considerable reduction of c-di-GMP binding.

had the most profound effect among the three phenotypes tested, followed by the mutation in L10. The mutation in L58 caused only minor changes in both motility and surface pilin production, and no difference in biofilm formation. There is a correlation between the ability of the mutants to complement the absence of wild-type MshE and their affinity for c-di-GMP *in vitro*. A mutation in Gly11 abrogated c-di-GMP binding, a mutation in Leu10 showed a 36-fold increase in the dissociation constant ($K_D$), while a mutation in Leu58 showed only a 10-fold increase in the $K_D$ (Supplementary Table 1). Altogether, the *in vitro* and

*in vivo* results suggest that binding of c-di-GMP is important for MshE activity; however, the affinity of the L58A mutant for c-di-GMP is still high enough to support biofilm formation.

We then analysed the biological consequences of multiple substitutions in L10, L54 and L58. As expected, when L10 and L58 were substituted with alanines the *mshE* variant was no longer able to complement Δ*mshE in trans*. When a mutation in Leu54 (L54A) accompanied mutations in either Leu58 (L54A-L58A) or Leu10 and Leu58 (L10A-L54A-L58A), the resulting construct was able to fully complement the pilus production

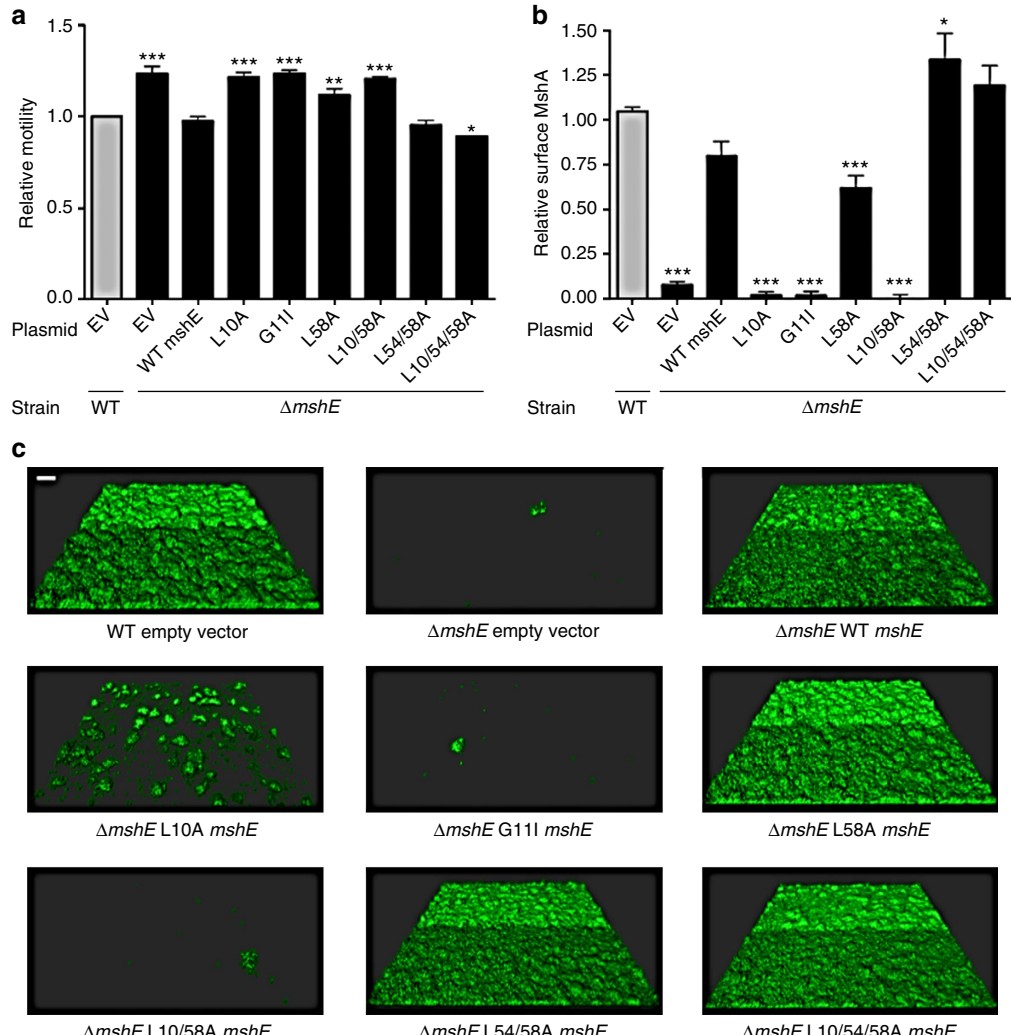

**Figure 4 | The ability of MshE to bind c-di-GMP affects MshA pilus production, biofilm formation and motility.** (**a**) Analysis of motility. (**b**) Analysis of surface MshA pilin production. MshA pili production was determined in WT or $\Delta mshE$ cells by ELISA in strains expressing wild-type or mutated versions of pMMB-*mshE*. In both **a** and **b**, the graph depicts mean and error bars representing s.d. The data were analysed with an one-way ANOVA followed by a Dunnett's Multiple Comparison test. (*$P \leq 0.05$, **$P \leq 0.01$, ***$P \leq 0.001$). EV indicates an empty vector, WT the wild type, other columns represent an *mshE* deletion mutant ($\Delta mshE$) complemented with plasmid-borne *mshE* genes with various amino-acid changes. (**c**) Analysis of biofilm formation. Three-dimensional biofilms of WT or $\Delta mshE$ strains expressing wild-type or mutated versions of *mshE* were grown for 24 h in a flow cell system. Images were acquired with confocal laser scanning microscopy. A representative image from each condition is shown.

defect, biofilm formation and the hypermotile phenotype of $\Delta mshE$ (Fig. 4). The simultaneous mutation of these three conserved Leu residues abrogates the ability of $Vc$MshEN to bind c-di-GMP *in vitro*, suggesting that this variant is constitutively active *in vivo* and may not require binding of c-di-GMP. Further investigation will be required to understand the underlying mechanisms by which c-di-GMP binding to MshEN controls the MshE activity (or assembly).

**MshEN as a ubiquitous regulatory domain.** Using the identified c-di-GMP-binding motif for sequence database searches retrieved numerous proteins with various versions of this motif, see an alignment of representative sequences on Fig. 2a and a sequence logo on Fig. 2b. In total, >10,000 proteins containing this domain, including the versions with the perfectly preserved c-di-GMP-binding motif, are found in representatives of many bacterial lineages (Supplementary Tables 3–5; Supplementary Fig. 3). Such a wide distribution identifies the MshEN domain as

an ancient bacterial regulatory module. The number of MshEN-encoding genes varies from 1 or 2 in most bacteria to as many as 22 in *Myxococcus xanthus* (Supplementary Table 3). Some of these proteins are encoded in typical *gsp* or *pil* operons, indicating that MshEN-containing ATPases are involved in both typical type II secretion and in formation of type IV pili (Supplementary Fig. 3a). In addition, MshEN is also found to associate with a variety of other domains, such as the two-component receiver (REC), c-di-GMP phosphodiesterase HD-GYP, chemotaxis protein kinase CheA and phosphatase CheX, adenylate/guanylate cyclase and Ser/Thr protein kinase (Supplementary Table 4; Supplementary Fig. 3b), indicating potential involvement of c-di-GMP in a variety of regulatory processes. MshEN domains are also found in certain organisms that do not encode any other (known) c-di-GMP receptors, partially resolving the enigma of c-di-GMP signalling in those bacteria (Supplementary Table 5). However, it is important to note that XpsE (or GspE) from Xcc17, a homologue of $Vc$MshEN, has lost the ability to bind c-di-GMP due to the changes in several key conserved residues (Fig. 2), such

as G11V, I19R, L29R, R38G, G40L, L54H, L58C and Q61V (Fig. 2a). Similar amino acid changes, likely causing a loss of c-di-GMP binding, could be seen in a number of MshEN domains (Supplementary Table 3).

Since c-di-GMP is bound primarily within the MshEN_N subdomain, the role of the C-terminal domain (MshEN_C, $VcMshE_{65-145}$) in c-di-GMP binding remained unclear. To explore this issue, we have generated a series of MshEN analogues with different MshEN_C subdomains and used ITC to check their binding affinities with c-di-GMP (Supplementary Figs 3b and 4). Intriguingly, while most MshEN analogues exhibited binding constants of ca. 0.5 μM, the middle domain of the PilB protein from *Deinococcus geothermalis* (GenBank accession ABF44566, see Supplementary Figs 3b and 4) had a $K_D$ of 13.5 nM (Supplementary Table 6). Other MshEN_N-containing constructs could also bind c-di-GMP with the affinity constant close to 0.5 μM (Supplementary Fig. 4). In fact, when the MshEN_N domain was separated from the MshEN_C domain, the isolated $VcMshEN_N{}_{1-62}$ domain bound c-di-GMP even stronger ($K_D$ 0.35 μM, Supplementary Fig. 4) than the complete $VcMshEN_{1-145}$ domain. Bioinformatics studies also show that the MshEN_N domain can form a $(MshEN_N)_2$ homodimer, or fuse with other domains with known or unknown functions (Supplementary Fig. 3b; Supplementary Table 4), with similar or stronger binding strengths. Finally, the binding affinities of several intact proteins with the MshE domain have also been checked. They all exhibited low-micromolar or stronger affinities with c-di-GMP (Supplementary Table 6; Supplementary Fig. 4).

## Discussion

The discovery of c-di-GMP[12] and environmentally regulated enzymes that participate in its turnover left the field with a well-known conundrum, referred to as an army with too many officers (those sending regulatory orders) and too few soldiers (c-di-GMP receptors that carry out those orders)[1]. This conundrum was only partially solved by the identification of the PilZ domain as a c-di-GMP adaptor protein[13]. It was clear that there must be other receptors[13]. In the past several years, some additional c-di-GMP receptors were identified[2], including the recently characterized biofilm regulator FleQ with an AAA$^+$-type ATPase domain[14]. Some common mechanisms of c-di-GMP binding have been delineated[2]. These include (1) side chain atoms of Arg or Glu/Asp are involved in specific interactions with the Hoogsteen-edge or Watson–Crick (WC)-edge atoms, respectively, of the guanine base; (2) aromatic amino acid residues are usually involved in stacking with the guanine base; (3) Arg residue can also use its side chain atoms to form a cation-π or CH-π interaction with the guanine base; (4) hydrophobic residues such as Leu/Ile/Val can also form CH-π interactions with the guanine base. Several c-di-GMP-binding motifs incorporating one or more of these principles have been identified, including the inhibitory RxxD motif in the GGDEF domain, ExLxR motif in the EAL domain, RxxxR and DxSxxG motifs in the PilZ domain, and the RxD-x$_8$-RxxD motif in the transcriptional regulator BldD[2]. However, the MshE and PA14_29490 sequences lack any of these canonical motifs and were expected to adopt a new c-di-GMP-binding mode. Indeed, Arg9 and Gln32, but not Asp12, were important for c-di-GMP binding in MshE[5] and the $^{55}LxxxLxxQ^{62}$-binding motif was identified in the *P. aeruginosa* protein PA3740 (ref. 15), which also contains the MshEN domain (Fig. 3a).

*Vc*MshE, a homologue of XpsE from *Xcc*17 (ref. 7) proved to be an efficient c-di-GMP binder[4,5]. The *Xc*XpsEN crystal structure has been solved a decade ago and found to crystallize in two different forms (closed versus open) depending on the precipitant conditions[6]. However, the protein phases of the *Vc*MshEN-c-di-GMP complex could not be determined by molecular replacement (MR) method using either the open or closed form of *Xc*XpsEN as a template, indicating certain differences in protein conformation between the c-di-GMP-binding *Vc*MshE and non-binding *Xc*XpsEN (see Fig. 1b). Yet, the most amazing finding in this report is that no side chain atom of conserved residues in the motif was involved in recognizing the guanine base via either its Hoogsteen- or WC-edge, as has been shown to be the case in all c-di-GMP-binding motifs for which the tertiary structures have been determined until now[2]. Only a non-conserved residue Asp108 in the MshEN_C domain was found to form a polar interaction with the WC-edged $2NH_2$ group of Gua2 (Fig. 2a), and even that interaction was dispensable for c-di-GMP binding (Supplementary Fig. 4). Instead, we found that polar interactions in the MshEN-c-di-GMP complex were achieved mainly by the two H-bonds from the backbone amide protons of the Gly11-Asp12 or Gly40-Asp41 dipeptide with the Hoogsteen-edge $O^6$ and $N^7$ atoms of each guanine base (Fig. 4a,b). Therefore the unique interactions of c-di-GMP with MshEN seem to be driven mainly by the three highly conserved hydrophobic Leu residues that form a triangular cluster to hold each guanine base via extensive CH-π bond interactions (Fig. 4b). The fist Arg residue is also involved in hydrophobic interaction with the guanine base from the other side, but not in H-bonding either (Fig. 4). This bizarre binding mode was confirmed by the ITC data of the $MshEN_{L58A}$ variant, in which alteration of one of the conserved Leu residues (Leu58) reduced its affinity towards c-di-GMP by > 10-fold (Supplementary Table 1), and variation of 2 or 3 Leu residues practically abolished the c-di-GMP binding, decreasing it to a level that is undetectable by ITC.

While the affinity of the MshEN domain towards c-di-GMP is higher than that of any c-di-GMP-binding protein reported to date, it is still lower than the affinity of the c-di-GMP-binding riboswitches. Two types of c-di-GMP riboswitches (GEMM I and GEMM II)[16], one type of c-di-AMP riboswitch[17–19], as well as a recently discovered cGAMP riboswitch[10,20], have been characterized. In *V. cholerae*, there are two class I c-di-GMP-binding riboswitches, Vc1 and Vc2, both of which have been experimentally characterized. However, the higher affinity of riboswitches to the c-di-GMP ligand as compared with the affinities of most c-di-GMP-binding proteins is a natural consequence of the entirely different mechanism of transcriptional regulation by these molecules.

Interestingly, c-di-GMP (and cGAMP) and c-di-AMP riboswitches bind their cognate ligands in entirely different modes, with c-di-GMP and cGAMP bound in stacked conformation[16,21], while bound c-di-AMP is in an extended-bulge conformation[17–19]. Since c-di-GMP in complex with *Vc*MshEN_N domain also adopts a similar extended shape, we have compared their binding schemes in more detail to learn whether the Nature has evolved protein and RNA to bind these two cyclic di-nucleotides in the same way (Supplementary Fig. 6). There are indeed similarities and differences between these two types of complexes. In both of them, the purine bases are sandwiched between two hydrophobic moieties, with their base Hoogsteen-edge hetero atoms involved in binding with the protein or RNA backbone atoms. However, the binding characteristics are rather distinct, especially those with the Hoogsteen-edge of purine base. In the c-di-AMP riboswitch, both adenine bases in the c-di-AMP ligand uses their $N^6$ (H-bonding donor) and $N^7$ (H-binding acceptor) atoms to form H-bonds with the lone pair of electrons and hydrogen atom of the U7-2′ hydroxyl group, or the G27-2′ and 3′-OH groups,

respectively[17–19] (Supplementary Fig. 6a). However, in the c-di-GMP-binding MshEN, both guanine bases of c-di-GMP use their guanine $N^7$ and $O^6$ atoms (both H-bonding acceptors) to form H-bonds with the amide protons of protein backbone Gly11 and Asp12 residues (see Supplementary Fig. 6 for a detailed comparison).

The MshEN domain was initially described in the ATPases involved in type II secretion and formation of type IV pili[4–6]. Now, using the 53-residue-long c-di-GMP-binding motif to search public databases allowed identification of many additional proteins as potential c-di-GMP receptors. A number of MshEN-containing proteins have been found in the genomes of widely used model bacterial organisms (Supplementary Table 3). Some of these proteins have been experimentally characterized, including those involved in swarming motility in *P. aeruginosa*[15], formation of the thick pili, cell motility and transformation competency in *Synechocystis* sp.[22], fruiting body formation in *M. xanthus*[23], bacteriophage N4 adsorption in *Escherichia coli*[24], and exopolysaccharide biosynthesis in *Methylobacillus* sp.[25] (Fig. 3a and Supplementary Table 3). However, it must be noted that the MshEN domain is often associated with previously uncharacterized domains. This is the case, for example, for *M. xanthus* protein FrgA, which is required for normal swarming and fruiting body formation in this bacterium[23], and the recently described swarming motility protein PA3740 from *P. aeruginosa*[15]. Characterization of MshEN as a c-di-GMP-binding regulatory domain will help in characterization of these additional domains and help in understanding the functions of the respective proteins. As discussed above, a perfect conservation of the c-di-GMP-binding motif may not be required for c-di-GMP binding. Thus, a recent work demonstrated tight binding of c-di-GMP to the MshEN-containing two-component response regulator Bd2402 of *Bdellovibrio bacteriovorus*[26], but the authors' screen also detected Bd1509 and Bd1596, two other MshEN-containing proteins of this bacterium with less conserved binding sites (Supplementary Table 3).

The exact mechanism of regulation by the MshEN domain remains obscure. It possibly works by increasing export of the MshA pili monomer outside the cell to form mature pili to increase bacterial attachment and biofilm formation. Similarly, c-di-GMP also enhances the activity of PA14_29490 to help export virulence factors via the T2SS pathway. In these cases, secretion of protein requires energy that can be supplied by the ATP hydrolysis catalysed by these critical secretion enzymes. However, the previous report showed that the ATPase activity of the full-length MshE protein was largely unaffected by the addition of external c-di-GMP (stimulation by <10%)[5]. These observations were confirmed in the course of this work, using the native MshE and its mutant variants. On the other hand, the *in vivo* biofilm formation by *V. cholerae* is highly sensitive to the presence of c-di-GMP, as well as to variations in the c-di-GMP-binding residues (Fig. 4). This inconsistency suggests that MshE requires some external factors to form an active ATPase complex. In fact, EpsE, another secretion ATPase from *V. cholerae*, has been reported to exhibit strong ATPase activity (stimulated >100-fold) only in the presence of an interacting partner, the inner membrane protein GspL, as well as phospholipids such as cardiolipin[27]. A similar effect has been observed for the XpsE ATPase from Xcc17 (refs 27,28). In our hands, MshE remained mostly monomeric in the solution; this inability to form a stable oligomer might be the reason for the dramatic difference in the MshE performance *in vitro* and *in vivo*. A bitopic protein such as GspL, which contains a cytoplasmic domain for binding ATPase, as well as a membrane helix to attach to phospholipid bilayer, may be required to form

an active secretion ATPase complex capable of secreting PilA or MshA monomer out of the cell for pilus formation[29]. However, no extra factors participating in the assembly of the active MshE complex have been characterized so far. Further efforts would be required to characterize such potential effectors and relate the effect of the presence of c-di-GMP and variation of c-di-GMP-binding residues with the MshE ATPase activity *in vitro*.

Bacteria have been found to use a remarkable array of sophisticated secretion systems to export various virulence factors across the bacterial cell envelope, and structural and molecular mechanisms of six secretion systems (types I–VI) have been described in Gram-negative bacteria[30]. Recently, c-di-GMP has also been found to play an important role in regulating secretion ATPases FliI, HrcN and ClpB2 involved in the flagella export, T3SS and T6SS[31], respectively. Yet, unlike MshE and PA14_29490, addition of c-di-GMP to these enzymes was found to inhibit, rather than enhance, their ATPase activities. FliI was proposed to form a hexamer, with the c-di-GMP-binding site, which includes two Arg and one Glu residues, situated between its two subunits[31]. The transcriptional regulator FleQ, whose structure has been recently published[14], is also different from MshE: in FleQ, c-di-GMP is bound by the ATPase domain itself, whereas in MshE the ATPase domain is not involved in c-di-GMP binding, as was clearly demonstrated before[5]. It therefore appears that bacteria adopt the ubiquitous c-di-GMP molecule as a master controller to regulate a variety of protein and polysaccharide secretion systems in distinct ways. It remains to be seen, however, whether these newly described secretion ATPases adopt the c-di-GMP-binding modes that are in any way similar to the one described in the present manuscript.

Structural characterization of the MshEN domain in one of the secretion ATPases here revealed a unique 53-residue-long c-di-GMP-binding motif that is widespread in the bacterial kingdom. Its wide phylogenetic distribution makes MshEN only the second, after PilZ, truly widespread c-di-GMP-responsive regulatory module. In many bacteria, it is the only (known) one (Supplementary Table 5). To the best of our knowledge, the guanine nucleotide-binding mode described in this work has not been reported before. Indeed, stacking interaction with the guanine base has been seen for aromatic residues such as Tyr, Trp, Phe or even His, while Leu was considered an unlikely residue to stack on the guanine base[32]. Yet in this case, there are two sets of three highly conserved Leu residues forming two triangles that stack on the two guanine bases of c-di-GMP (Fig. 3). Such hydrophobic interactions seem to be quite important, and a single change in any of these residues causes dramatic change in c-di-GMP binding and bacterial biofilm formation. It remains to be seen whether a similar structural arrangement might be present in other proteins and/or be used for synthetic biology projects.

## Methods

**Sample preparation.** The MshEN domain and MshEN_N domain containing gene sequences were purchased from GENEWIZ (Plainfield, NJ, USA), with codons optimized for *E. coli* expression. The gene sequences were PCR-amplified using the *Pfu*Ultra DNA polymerase (Stratagene) and digested with restriction enzyme SSPI before cloning into the pET21a vector vectors using primers listed in the Supplementary Table 7. A ligation-independent cloning approach was carried out to obtain the desired constructs[33]. A series of substitutions of amino-acid residues in the MshEN domain were carried out by the Quick Change site-directed mutagenesis method (Stratagene) using the *Pfu*Ultra DNA polymerase. Appropriate mutations were further confirmed by DNA sequencing. All vectors were transformed into *E. coli* BL21 (DE3) host cells and grown in lysogeny broth (LB) medium at 37 °C until an $OD_{600}$ of 0.6 was attained. Overexpression of the His$_6$-tagged GST-fused target protein was induced by addition of 0.5 mM isopropyl β-D-1-thiogalactopyranoside (IPTG) at 289 K for 20 h. The cells were harvested, re-suspended in equilibration buffer (20 mM Tris–HCl pH 8.0 and 80 mM NaCl)

and lysed using a microfluidizer (Microfluidics). After centrifugation, the target protein was purified by immobilized metal-affinity chromatography (IMAC) on a nickel column (Sigma) and eluted with a 20–300 mM imidazole gradient in 20 mM Tris–HCl pH 8.0 and 80 mM NaCl. The fractions containing VcMshEN were monitored by SDS–PAGE, recombined and dialyzed repeatedly against 20 mM Tris–HCl pH 8.0 and 80 mM NaCl. After buffer exchange, the His6 tag and GST fusion protein were cleaved from VcMshEN by tobacco etch virus protease at 16 °C for 16 h and removed by IMAC on a nickel column. The final protein had >98% purity and contained a non-native tripeptide (Ser-Asn-Ala) followed by the target protein sequence.

C-di-GMP was produced by an enzymatic method using an altered thermophilic diguanylate cyclase[34], or purchased from the BioLog company (Berlin, Germany). C-di-AMP was generated by an efficient enzymatic method using a diadenylate cyclase DisA from *Bacillus thuringiensis* BMB 171 (btDisA)[35]; cGAMP was purchased from the BioLog company.

**Crystallization of the MshEN/c-di-GMP complex.** Crystallization of the native MshEN/c-di-GMP complex was performed at 295 K using the vapour-diffusion method. Crystals with a diamond-shaped morphology were obtained after 1 week by mixing protein solution (8 mg ml$^{-1}$) with precipitant solution comprising 0.1 M sodium cacodylate pH 6.5, 0.2 M magnesium acetate, and 20% (w/v) PEG 8000. Before flash-cooling in liquid nitrogen, the crystals were briefly washed in a solution consisting of 0.1 M sodium cacodylate pH 6.5, 0.2 M magnesium acetate and 30% (w/v) PEG 8,000 for cryoprotection.

For phasing, crystals of MshEN/c-di-GMP complex in its Se-labelled form were obtained after 10 days by mixing protein solution at 7.5 mg ml$^{-1}$ with precipitant solution consisting of 1.6 M sodium citrate tribasic dihydrate pH 6.5. The crystals were briefly rinsed in cryoprotectant solution comprising the corresponding precipitant solution supplemented with 20% (v/v) glycerol before flash-cooling in liquid nitrogen.

**X-ray data collection and structure refinement.** The selenium single-wavelength anomalous dispersion (Se-SAD) phasing method was used to determine the phasing of the MshEN − c-di-GMP complex using data collected at the peak Se wavelength (0.97902 Å) at 100 K using the 15A1 beamline of National Synchrotron Radiation Research Center (NSRRC) in Taiwan.. The native data set was collected at wavelength of 1.00000 Å at 100 K using the SP44XU beamline of SPring-8 Synchrotron Facility in Japan. The data were indexed and integrated using the HKL2000 processing software[36], generating data set that were ∼91.2% complete with overall $R_{merge}$ of 3.5–3.8% on intensities. Experimental phases were calculated using PHENIX program with three heavy atoms identified by AutoSol[37]. The quality of the electron-density map obtained after solvent flattering enabled the auto-building of an initial model using *phenix*.AutoBuild[38]. This model was further optimized by iterative cycles of manual building using the graphics programme *MIFit* and refinement package implemented in the *phenix.refine* and was further used to determine phases for the high-resolution native data sets by MR using *phenix.automr*. The resulting models were further refined by alternating rounds of manual rebuilding and refinement as described above. The maximum likelihood coefficient is 0.19 Å based on sigma-A value of 0.93 (0.88) and the Wilson B factor (16.5 Å$^2$). The CC$^{1/2}$ values are 99.2 (81.6). The Ramachandran plot showed that the protein backbone torsional angles were all located in the most favourable regions (98.55%) or additionally allowed regions (1.45%), with no residue located in generously allowed region. The data collection and refinement statistics are summarized in Table 1.

**ITC measurements of MshEN with various ligands.** The samples of MshEN for ITC were extensively dialyzed against the assay buffer (80 mM NaCl, 20 mM Tris pH 8.0). The protein samples were first diluted with the assay buffer to 50 μM before loading into the ITC cell. C-di-GMP was diluted in the same way to 1.0 mM before loading into the syringe. Two microlitres of c-di-GMP solution were then injected into the cell at 3-min intervals. The ITC experiments with wild-type MshEN and its variants were performed in a VP-ITC (MicroCal, Northampton, MA, USA) and carried out at 25 °C, with the data fitted using the commercial Origin 7.0 program to obtain the ΔH and $K_D$ values[39]. The ITC measurement data for the MshEN domain and its variants, as well as MshEN-domain-containing proteins with c-di-GMP were listed in Supplementary Tables 1 and 6, respectively.

The ITC experiments on c-di-AMP and cGAMP binding to wild-type MshEN were carried out in a similar way.

**DSF of wild-type MshEN and its variants.** The DSF experiment[40] was carried out on a BioRad qPCR instrument in a buffer comprising 5 μl of 25 mM Tris pH 7.5, 100 mM NaCl and 0.2 mM MgCl$_2$, 18 μl of 1:1,000 dilution of SYPRO Orange Dye and 2 μl of 10 μM proteins in the presence or absence of 1 mM nucleotides in each well. The fluorescence was monitored when temperature was gradually raised from 20 to 90 °C in 0.5-°C increments at 30-s intervals. Melt curve data were plotted with Boltzmann model to obtain the temperature midpoint of unfolding of the protein using Prism 5.0 software (GraphPad).

**Strains and growth conditions.** The *E. coli* and *V. cholerae* strains used in this work are listed in Supplementary Table 8. The strains were grown on LB (1% tryptone, 0.5% yeast extract, 1% NaCl), pH 7.5 at 37 °C and 30 °C, respectively. LB agar medium contained 1.5% (w/v) of granulated agar (Difco). LB soft agar medium contained 0.3% (w/v) of granulated agar. The following antibiotics were added at the indicated concentration: ampicillin at 100 μg ml$^{-1}$ or 20 μg ml$^{-1}$, and rifampicin at 100 μg ml$^{-1}$. When indicated, IPTG was added to a final concentration of 0.1 mM.

**Site-directed mutagenesis of mshE.** Point mutations to *mshE* to change the amino acids predicted to be important for c-di-GMP binding by MshEN were introduced using the Q5 site-directed mutagenesis kit from New England Biolabs (Ipswich, MA) following the manufacturer's instructions. The plasmid pVL393-*mshE*[5], containing the wild-type *mshE* structural gene, was used as a template to introduce point mutations at positions 10 (L10A), 11 (G11I), 58 (L58A) and 54–58 (L54A–L58A). The resulting plasmids pVL393-*mshE*-L58A and pVL393-*mshE*-L54A-L58A were used as templates to introduce an additional point mutation at position 10, resulting in constructs pVL393-*mshE*-L10A-L58A and VL393-*mshE*-L10A-L54A-L58A, respectively. The primers used to introduce these point mutations were designed using the NEBaseChanger interactive tool from New England Biolabs. All constructs were analysed through DNA sequencing to confirm the presence of the desired point mutations and to verify that no additional mutations were generated. All constructs were mobilized into ΔmshE/ Tn7::gfp through biparental mating using as donor *E. coli* S17-1 λ *pir* harbouring the desired complementation constructs. Transconjugants were selected on plates containing LB agar medium with rifampicin and ampicilin (100 μg ml$^{-1}$).

**Surface Pilin ELISA.** Surface pili composed of MshA were quantified using an ELISA as previously described. Briefly, overnight culture was diluted 1:100 in fresh LB medium and grown to OD600 0.5 at 30 °C with 100 μM IPTG and 100 μg ml$^{-1}$. Cells (125 μl) were added to a 96-well plate (Greiner Bio-One, Monroe, NC) and incubated at 30 °C for 1 h. Cells were fixed with 100 μl of methanol for 10 min at room temperature, and then washed twice with PBS. Samples were blocked in 5% nonfat dry milk and immunoblotted with polyclonal rabbit anti-MshA (1:1,000 dilution, gift of J. Zhu, U. Penn.) and horseradish peroxidase-conjugated secondary antibody (Santa Cruz Bio-technology, Santa Cruz, CA). After three washes in PBS, 100 μl of TMB (eBioscience, San Diego, CA) was added and incubated for 30 min at room temperature followed by the addition of 100 μl of 2 N H2SO4. Absorbance was recorded at 490 nm and the samples were normalized to the change in WT. Two biological replicates were assayed in triplicate and statistical significance was determined with an one-way analysis of variance (ANOVA) followed by a Dunnett's Multiple Comparison test.

**Analysis of biofilm formation.** Overnight cultures were diluted 1:20 in LB containing 100 μM IPTG and 100 μg ml$^{-1}$ ampicillin and grown at 30 °C for 1 h. Inoculation of flow cells was done by diluting overnight-grown cultures to an OD600 of 0.04 and injecting into a μ-Slide VI0.4 (Ibidi, Martinsried, Germany). Cells were allowed to adhere to the surface at room temperature for 1 h. Flow of 2% v/v LB (0.02% tryptone, 0.01% yeast extract, 1% NaCl; pH 7.5) was initiated at a rate of 7.5 ml h$^{-1}$ and continued for 24 h. Confocal images were obtained on an Olympus FV1,000 Confocal Laser Scanning microscope (CLSM). Images were obtained with a × 40 oil-immersion objective and were processed using Imaris (Bitplane, Zurich, Switzerland). Quantitative analyses were performed using the Comstat2 software package[41,42]. Statistical significance was determined using one-way ANOVA with Dunnett's Multiple Comparison test. Biofilms were imaged in triplicate. Images presented are from one representative experiment.

**Motility in soft agar plates.** The motility of different *V. cholerae* strains was determined by measuring the zone of migration of cells growing in plates containing LB soft agar plus ampicillin (20 μg ml$^{-1}$) and IPTG (0.1 mM). Three independent colonies from overnight LB agar plates containing ampicillin (100 μg ml$^{-1}$) were transferred to LB soft agar plates and left to grow for 16 h at 30 °C. The motility diameter was measured and normalized to the average of three independent colonies of the parental strain on each plate. The data were analysed with a one-way ANOVA followed by a Dunnett's Multiple Comparison test. *$P \leq 0.05$, **$P \leq 0.01$, ***$P \leq 0.001$.

**Bioinformatic analyses.** Iterative sequence database searches against NCBI and EBI protein databases were initially performed using PSI-BLAST[43] and JackHMMer[44], respectively, using the VcMshEN$_{1-145}$ sequence as query. Subsequent searches used the sequences of cce_1044 (GenBank accession ACB50395) and proteins listed in Supplementary Table 5 as queries. Domain architectures of the retrieved proteins were analysed using the CDD[45] and InterPro[46] databases. MshEN domain counts in individual bacterial genomes were obtained by searching species- or taxon-specific databases using PSI-BLAST[43] and/or PHI-BLAST[47] with a relaxed 24-residue motif [RKPN]hGxxhhxx[GAS]hhxxxxhxxxhxx[QNKRYWST], where x stands for any amino acid and h indicates a hydrophobic amino acid residue (LIVAFYWM); the retrieved sequences were manually checked and the likelihood of

c-di-GMP binding was evaluated on case-by-case basis. Domain assignments were checked using HHPred[48]. The sequence logo was constructed from an alignment of protein sequences retrieved by PSI-BLAST searching of the $VcMshEN_{1-70}$ against the NCBI non-redundant protein database, followed by filtering for the presence of four Gly residues, corresponding to Gly11, Gly18, Gly40 and Gly47 (VcMshEN numbering). Extra columns that were absent in VcMshEN were deleted and the entire alignment was filtered to remove duplicate sequence fragments. The resulting alignment of 2,021 non-identical sequences was used to build a sequence logo using the WebLogo tool[49].

The search for other structures with a nucleotide-binding mode based primarily on hydrophobic interactions was performed using the three-dimensional Environment service at the EBI's PDBeMOTIF[50] web site with additional searches against the Nucleotide binding database NBDB[51].

**Data availability.** All relevant data are available from the authors.

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

## Acknowledgements

This work was supported in part by the Ministry of Education, Taiwan, ROC under the ATU plan, and by the Ministry of Science and Technology, Taiwan, ROC (102-2113-M-005 -006 -MY3) to S.-H.C. and NIH award RO1 AI102584 to F.H.Y. M.Y.G. was supported by the NIH Intramural Research Program at the US National Library of Medicine. We would also like to thank the National Synchrotron Radiation Research Center (NSRRC) in Taiwan, and the SPring-8 Synchrotron facility in Japan for assistance of the X-ray data collections.

## Author contributions

Y.-C.W., K.-H.C., Z.-L.T., J.H., C.J.J. and D.Z.S. performed the experiments; Y.-C.W. and K.-H.C. performed structure determination; Y.-C.W., K.-H.C., F.H.Y., M.Y.G. and S.-H.C. analysed the data; Y.-C.W., C.J.J., D.Z.S., M.Y.G. and S.-H.C. prepared the figures; F.H.Y., M.Y.G. and S.-H.C. wrote the manuscript together with comments from all authors; S.-H.C. supervised the work.

## Additional information

**Accession codes:** The structure factors and coordinates of the MshEN-c-di-GMP complex have been deposited in the Protein Data Bank with accession number 5HTL.

**Competing financial interests:** The authors declare no competing financial interests.

