## [Peer Review File · Nature Communications]

Reviewer #1 (Remarks to the Author):

I had previously reviewed this manuscript, and all my concerns have been addressed. One minor point, in page 4, it should be "four asymmetric units combine together to form a unit cell comprising the P212121 space group" not "several asymmetric units)

Reviewer #2 (Remarks to the Author):

In general, the response to review is measured and sufficient. Most of my direct comments have been addressed, with the following caveats:

-I still recommend the removal of stereo and point to its general disuse in recent times.

-The comment "But in either case, the stabilities of the

protein variants will be reflected by their different binding affinities with their ligands" is only partly true - stability and affinity are not always clearly related/dissectable, and it would benefit the study to measure stability of the mutants more directly.

-The added CC1/2 information is welcomed, but I still think with an I/sigI around 2 and CC1/2 >> 50, there is still an underestimate of their effective resolution.

-Could the authors note (even if just in correspondence) whether there is any effect in diversifying their PDB motif search to include V/I/M/F as well as Leu?

-The new inclusion of spacegroup information on page 4 is rather non-standard; P212121 is not written in the correct format, and the statement that the spacegroup arises via inclusion of several ASUs is a truism and not required

-The introductory sentence on p.16 about the FleQ structure is nonsensical and needs to be edited.

Reviewer #3 (Remarks to the Author):

This is the revised and resubmitted version of a manuscript previously submitted to Nature.

I still find the topic exciting and the manuscript improved. As the authors promise in their answer to provide data on ATPase activity of MshE proteins with relevant mutations in the MshEN domain in a revised version, I am looking forward to this complementation. In addition, as *Vibrio cholerae* is not known to produce c-di-AMP, but c-GAMP, the authors should, at least confidentially to the reviewers, report the data of c-GAMP binding to the MshEN domain. The discussion about the differences in the binding modes of cyclic di-nucleotides of RNA riboswitches versus the MshEN domain is quite long and can be, at least with respect to the details, moved to a Supplementary description.

Minor comments

p.2, l.4: ATPases (MshE)?

p.3, l.20: 53-residue motif and p.3, l.21: novel c-di-GMP motif. In other places in the manuscript the 53-residue motif is called motif, which I can agree with. The authors use domain and subdomain in another context.

p.3, l.17: Why problem?, maybe 'enigma'

p.14, l.14: ...mother nature... maybe replace by a less emotional expression

Supplementary Table 9: are these representative data from one experiment? Biological and technical replicates are not clearly reported.

There are some typos in the manuscript on several occasions. E.g. Abstract: p.2, l.16: Bioinformatic analyses; p.14, l.1: ...in more detail...; p.16, l.10/11: sentence not complete; Supplementary Figure 6: ...red dotted lines.

Reviewer #1 (Remarks to the Author):

I had previously reviewed this manuscript, and all my concerns have been addressed.

One minor point, in page 4, it should be "four asymmetric units combine together to form a unit cell comprising the $P2_12_12_1$ space group" not "several asymmetric units).

AU: A similar concern has been raised by Reviewer 2. In this revised version, this sentence has been deleted.

Reviewer #2 (Remarks to the Author):

In general, the response to review is measured and sufficient. Most of my direct comments have been addressed, with the following caveats:

I still recommend the removal of stereo and point to its general disuse in recent times.

AU: As suggested, we have removed the stereo from Fig. 3a and Fig. 3c, but we prefer to keep the stereo rendering in Fig. 2c, due to its complex nature.

The comment "But in either case, the stabilities of the protein variants will be reflected by their different binding affinities with their ligands" is only partly true - stability and affinity are not always clearly related/dissectable, and it would benefit the study to measure stability of the mutants more directly.

AU: As suggested, we have also checked the stabilities of native MshEN_N and its representative active site variants by using the DSF method in the absence or presence of c-di-GMP (Supplementary Table 2). The wild type and all tested representative variants had their T_m values in the 46 – 60 ° C range, indicating that these variants are stable in the absence of c-di-GMP. For the G40I and G11L variants, the presence of c-di-GMP did not change their T_m values, consistent with the poor, if any, binding of the ligand (see Supplementary Table 2). For single (L10A and L25A) or multiple (L54A/L58A, or L10A/L54A/L58A) replacements that preserve the hydrophobic character of the active site residues, the T_m values increased by 5° or more in the presence of c-di-GMP, indicating that these variants could still be stabilized by c-di-GMP binding.

The added CC1/2 information is welcomed, but I still think with an I/sigI around 2 and CC1/2 >> 50, there is still an underestimate of their effective resolution.

AU: The structure of MshEN/c-di-GMP complex has been solved to a high resolution of 1.37 Å. Thus we think this is good enough to reveal the unique interactions between c-di-GMP and the surrounding active site residues of MshEN_N domain. Given our goal of characterizing the new c-di-GMP receptor and its binding mode, we do not believe that further improvements in resolution would significantly affect the conclusions of this work.

Could the authors note (even if just in correspondence) whether there is any effect in diversifying their PDBemotif search to include V/I/M/F as well as Leu?

AU: Our initial searches with PDBeMOTIF actually looked for Leu, Val, Ile and Arg residues using the [LIV][LIV][LIV]R pattern with guanine, adenine and various nucleotides as ligands. The retrieved hits (4qg4 and other SAMHD1 structures, 1p9b and other adenylosuccinate synthetases, 2qrd, 1uky, 2cdn and related kinases, and so on) were opened in PDBsum, the respective 'List of interactions' links were downloaded and analyzed one-by-one. We could not find any case where aliphatic residues formed a hydrophobic surface that would stack with a nucleobase. In most cases, retrieved interactions of LIV residues involved main chain nitrogen and oxygen atoms. We then polled our colleagues who are maintaining protein structure databases in NCBI and around the world asking them for such examples. All we heard back were suggestions to look at 1iyb, 4z0g, and other structures that involved stacking by aromatic residues. We are therefore quite confident that the described Leu-Leu-Leu arrangement represents a new kind of nucleotide binding mode; that is why we put it in the title of the paper. At the reviewers request, we tried the [LIVMF][LIVMF][LIVMF]R pattern. It returned several new structures, including and fructose-2,6-bisphosphatase 1c80 and xanthine oxidase 3nvw but again, no structures that we have checked exhibited anything similar to the Leu-Leu-Leu structural motif.

The new inclusion of space group information on page 4 is rather non-standard; P212121 is not written in the correct format, and the statement that the space group arises via inclusion of several ASUs is a truism and not required.

AU: This sentence has been deleted.

The introductory sentence on p.16 about the FleQ structure is nonsensical and needs to be edited.

AU: The description of the FleQ structure in page 16 has been simplified to: “The transcriptional regulator FleQ, whose structure has been recently published, is also different from MshE: in FleQ, c-di-GMP is bound by the ATPase domain itself, whereas in MshE, the ATPase domain is not involved in c-di-GMP binding, as was clearly demonstrated before⁵”.

Reviewer #3 (Remarks to the Author):

This is the revised and resubmitted version of a manuscript previously submitted to Nature. I still find the topic exciting and the manuscript improved.

AU: Thank you! We greatly appreciate the encouragement.

As the authors promise in their answer to provide data on ATPase activity of MshE proteins with relevant mutations in the MshEN domain in a revised version, I am looking forward to this complementation.

AU: We have indeed checked the ATPase activities of full-length MshE protein and some of its active site variants (data not shown). Consistent with the previous report¹, the activity of native MshE could only be enhanced somewhat (less than 10%) by the addition of external c-di-GMP. Our ATPase activity assays showed that the ATPase activities of native MshE and its variants are relatively low and are not responsive to the presence of c-di-GMP (data not shown). However, the *in vivo* biofilm formation by *V. cholerae* is highly sensitive to the presence of c-di-GMP as well as to variations in the c-di-GMP binding residue (Fig. 4). This discrepancy suggests that MshE requires some external factors to form an active ATPase complex. In fact, EpsE, another secretion ATPase from *V. cholerae*, has been reported to exhibit strong ATPase activity (stimulated more than 100-fold) only in the presence of an interacting partner, the inner membrane protein EspL, as well as the presence of phospholipids such as cardiolipin². A similar behaviour has been observed for GspE from *V. vulnificus* and XpsE from *X. campestris*. Thus the difference in the MshE functioning *in vitro* and *in vivo* could be due to the inability of the monomeric MshE to form a stable oligomer in solution (as observed by gel filtration chromatography, data not shown). A membrane protein such as EspL, which contains a cytoplasmic domain for binding ATPase and a membrane helix to attach to the phospholipid bilayer, may be required to form an active secretion ATPase complex for secreting PilA or MshA monomer out of the cell for pilus formation³. However, such extra factors for forming the active MshE complex are currently unknown, and more efforts would be required to characterize these effectors and relate the effect of the presence of c-di-GMP and variation of c-di-GMP binding residues with the MshE ATPase activity *in vitro*.

In addition, as *Vibrio cholerae* is not known to produce c-di-AMP, but c-GAMP, the authors should, at least confidentially to the reviewers, report the data of c-GAMP binding to the MshEN domain.

AU: We thank the reviewer for pointing out this important issue. We have tested the c-GAMP binding with MshEN by ITC and included these data in the manuscript as Fig. 1h. Indeed, we did detect weak to moderate binding ($K_D = 3.3 \times 10^{-4}$) of cGAMP with MshEN. This binding affinity is approximately 1000-fold less than that with c-di-GMP ($K_D = 5.0 \times 10^{-7}$) but is definitely better than that with c-di-AMP (non-detectable). We conclude that only guanine base can properly fit into the MshEN nucleotide binding pocket. That said, in the conditions when c-di-GMP is absent and c-GAMP is abundant (as described by Davis *et al.*, 2012⁴, for *V. cholerae* entering the host intestinal environment), binding of c-GAMP to MshE might be physiologically relevant.

The discussion about the differences in the binding modes of cyclic di-nucleotides of RNA riboswitches versus the MshEN domain is quite long and can be, at least with respect to the details, moved to a Supplementary description.

AU: We agree and have significantly shortened the comparison of the interactions of cyclic di-nucleotides with the RNA and protein receptors (top of page 14). Some paragraphs have been moved to the supplementary material (Supplementary Fig. 6).

Minor comments

p.2, 1.4: ATPases (MshE)?

AU: We have changed the phrase to “two ATPases (*Vibrio cholerae* MshE and its homolog)”.

p.3, 1.20: 53-residue motif and p.3, 1.21: novel c-di-GMP motif. In other places in the manuscript the 53-residue motif is called motif, which I can agree with. The authors use domain and subdomain in another context.

AU: The motif and domain are difficult to differentiate in the present case, as the mentioned 53-residue motif in fact forms a separate domain to bind c-di-GMP. But this 53-residue domain contains many non-conserved interrupting residues, so we now refer to the conserved residues as a motif, and to the whole 53-residue sequence as domain to make it clearer.

p.3, 1.17: Why problem?, maybe 'enigma'

AU: As suggested, we have changed the word to “enigma”.

p.14, 1.14: ...mother nature... maybe replaced by a less emotional expression

AU: We have changed the word to simply “nature”.

Supplementary Table 9: are these representative data from one experiment? Biological and technical replicates are not clearly reported.

AU: The data presented in Supplementary Table 9 were from three independent biofilms that were imaged in duplicate. This statement has been appended into the Table 9 legend.

There are some typos in the manuscript on several occasions. E.g. Abstract: p.2, l.16: Bioinformatic analyses; p.14, l.1: ...in more detail...; p.16, l.10/11: sentence not complete; Supplementary Figure 6: ...red dotted lines.

AU: The manuscript has been carefully checked to remove typos.

References:

- 1 Roelofs, K. G. *et al.* Systematic identification of cyclic-di-GMP binding proteins in *Vibrio cholerae* reveals a novel class of cyclic-di-GMP-binding ATPases associated with type II secretion systems. *PLoS Pathogens*, e1005232 (2015).
- 2 Camberg, J. L. *et al.* Synergistic stimulation of EpsE ATP hydrolysis by EpsL and acidic phospholipids. *EMBO J.* **26**, 19-27 (2007).
- 3 Korotkov, K., Sandkvist, M. & Hol, W. The type II secretion system: biogenesis, molecular architecture and mechanism. *Nat Rev Microbiol.* **10**, 336-351. (2012).
- 4 Davies, B. W., Bogard, R. W., Young, T. S. & Mekalanos, J. J. Coordinated regulation of accessory genetic elements produces cyclic di-nucleotides for *V. cholerae* Virulence. *Cell* **149**, 358-370 (2012).

Reviewer #2 (Remarks to the Author):

All my comments have been satisfactorily addressed. I look forward to seeing the manuscript in press.

Reviewer #3 (Remarks to the Author):

This is the revised version of a previously submitted manuscript. I have no more comments to make.

Referee #1 (Remarks to the Author):

The manuscript by Wang et al. describes the first crystal structure and structure-based mutagenesis of a recently discovered protein domain that recognizes the bacterial second messenger c-di-GMP. C-di-GMP has emerged as an universal player involved in bacterial "lifestyle" decisions, but based on phylogenetic analyses, primarily, it has been suspected that not all macromolecules that respond to this second messenger have been characterized. Last year, two groups identified members of a likely structurally novel class of c-di-GMP receptor proteins (references 2 and 3 in this MS), for which now Wang et al. present the near-atomic resolution crystal structure. With the structure of the domain in hand, the authors cast a bioinformatic net that demonstrates that this ~150 residue module is widely distributed across bacterial phyla. The work presented appears to be technically sound, and will have considerable impact among those interested in c-di-GMP biology. However, I feel that the

appeal of this work is somewhat narrow, since several other types of c-di-GMP (and other cyclic dinucleotide) receptors (both protein and RNA) are known, and since c-di-GMP signaling itself is now widely appreciated, even if many of its details are still being worked out. Thus, my recommendation is that this work is more appropriate for a specialist structural or molecular biology journal. I have one substantive and some minor suggestions.

1. An interesting feature of c-di-GMP recognition revealed by the Wang et al. is that the MshEN domain makes base-specific interactions with one of the nucleobases (Gua1) of c-di-GMP (using two consecutive protein backbone amides), but not with the other (Gua2), which is surrounded by a series of conserved hydrophobic residues. The mechanism of recognition of Gua1 precludes binding of an adenine residue at that location, and, consistent with that, the authors show through ITC that c-di-AMP does not bind to MshEN. However, the Gua2 binding site may be more permissive. Thus, I feel it is important to establish whether c-GMP-AMP, a relatively recently recognized bacterial (and eukaryotic) second messenger, can be bound by MshEN. If that were the case, the significance of the manuscript would be considerably greater. (I note that there are two recognized variants of this mixed cyclic messenger, which differ in the backbone stereochemistry. One of them is thought to be strictly eukaryotic, but it would be worth trying both).

AU: We thank the reviewer for these very helpful comments. But first we need to clarify that both Gua1 and Gua2 bases of c-di-GMP in complex with MshEN_N are recognized by the well conserved residues in a very similar way. Perhaps we did not describe that clearly enough owing to the text limitation in the first version of the manuscript. To make it clearer, we have included a new supplementary figure (Fig. 5) to compare the locations of the two guanine bases. From the figure, one can clearly see that, although the two guanine bases are located in a different environment (Fig. 2a and 2c), the C α ' atoms of the conserved binding residues are well aligned, with a RMSD of only 0.8 Å over all 25 residues. The α 1-linker- α 2 segment superimposes with the entire α 3-linker- α 4 segment, with most of the side chain atoms of the conserved residues including those from Leu10/Gly11/Asp12/Leu25/Gln32 situating almost on top of each other with Leu39/Gly40/Asp41/Leu54/Gln61. Only the side chains of the Arg9/Arg38 and Leu29/Leu58 residues exhibit some differences. Thus, it is unlikely that the Gua2 binding site will be more permissive. However, one cannot exclude the possibility that the MshEN_N domain may adjust its conformation to accommodate a cGAMP molecule, which has been found to be present in the *Vibrio cholerae*¹. We are planning to explore this possibility as soon as we get hold of the 3'-5' cGAMP compound, and will use ITC to check the binding constant. We will include this data in the next revised manuscript.

It has also been discovered recently that cGAMP can exist in the 3'-5' and 2'-5' configurations. The second 2'-5' form of cGAMP is produced only in the metazoan cells to induce the innate immunity response, and the cGAS enzyme is only active in the presence of foreign DNA^{2,3}. However, the 3'-5' form is present in prokaryotic cells and is produced by a constitutively active cGAS even in the absence of foreign DNA¹. Thus the 2'-5' cGAMP is less likely to interact with the MshEN_N domain, and we do believe that this experiment is not justified at this time.

2. The authors completely ignore the fact that there are two known structurally distinct classes of bacterial RNAs (riboswitches) that recognize c-di-GMP and one that binds c-di-AMP. Indeed, superficially at least, it looks like the MshEN-bound conformation of c-di-GMP may be rather similar to the riboswitch-bound conformation of c-di-AMP. Also, for the c-di-AMP riboswitch, it appears that steric exclusion is part of the mechanism for discriminating against c-di-GMP, and that may be relevant in thinking about the Gua2 site in the MshEN domain.

AU: We are certainly aware of the c-di-GMP-binding riboswitches, but due to the text limitation, we could only delete them in the first version of the manuscript. There are two class I c-di-GMP binding riboswitches, Vc1 and Vc2 in *Vibrio cholerae*, both have been experimentally characterized and are listed on the c-di-GMP census web site http://www.ncbi.nlm.nih.gov/Complete_Genomes/c-di-GMP.html, which is maintained by one of the authors (MYG). However, we would like to stress that (i) The much higher affinity of riboswitches to the c-di-GMP ligand as compared to the affinities of most c-di-GMP-binding proteins is a natural consequence of the entirely different mechanism of transcriptional regulation by these molecules; (ii) Among c-di-GMP-binding proteins, MshEN-containing proteins demonstrate some of the highest affinities toward their ligand (down to 15 nM), and (iii) While our Supplementary Table 5 did not include riboswitches, the above-mentioned web site includes riboswitch data from Rfam and clearly shows that most organisms listed in that Table do not encode any known (or predicted) riboswitches. Therefore, our statement that c-di-GMP binding by MshEN domains helps resolve the “too many officers, too few soldiers” conundrum still remains valid. Anyway, more descriptions about these strong c-di-GMP binding riboswitches will be appended in the main texts of the revised manuscript.

That said, we thank this reviewer for the excellent comment about the potential similarity between the conformations of c-di-GMP in complex with MshEN_N and c-di-AMP in complex with the riboswitch. Indeed, c-di-GMP and c-di-AMP both adopt a bulge-type extended conformation, with the two bases almost perpendicular to each other. To solidify this interesting result, we have made another Supplementary Fig. 6 to compare their different binding schemes. There are indeed similarities and differences between these two types of complexes. In both cases, the purine bases are sandwiched between two hydrophobic moieties, with their base Hoogsteen-edge hetero atoms involved in binding with the protein or RNA backbone atoms. However, the binding characteristics are rather distinct. In the protein-binding mode (shown in Fig. 3a), the first two helices of the four-helical bundle (α 1-linker- α 2) form a similar guanine binding environment with the last two helices (α 3-linker- α 4) to form a pseudo-dimer for binding Gua1 and Gua2 bases of the single c-di-GMP ligand (Supplementary Fig. 5). In contrast, in the RNA-based c-di-AMP riboswitches, each forms a pseudo-dimer to bind two c-di-AMPs. One adenine bases in the first c-di-AMP ligand (A1 α) uses their N6 (H-bonding donor) and N7 (H-binding acceptor) atoms to form H-bonds with the lone pair electrons and the hydrogen atom of the U7-2' hydroxyl group, respectively (Supplementary Fig. 6a). An additional H-bond between the N6 atom with the lone pair electrons of the U7-3' hydroxyl group is also observed⁴⁻⁶. Another adenine bases in the first c-di-AMP ligand (A1 β) also uses its Hoogsteen-edge N6 and N7 atoms to form similar H-

bonds with the G27-2' and 3'-OH groups (Supplementary Fig. 6a). But unlike the stacking of the guanine bases with the Arg side chain atoms and the surrounding tri-Leu hydrophobic cluster in the MshEN_N-c-di-GMP complex, the adenine base of c-di-AMP in this riboswitch is stacked by an adenine base (A9 or A100) in one side, and by a ribose of G81 or G61 from the other side (Supplementary Fig. 6a). The binding schemes of the second c-di-AMP ligand is similar to those of the first ligand and is shown in Supplementary Fig. 6b. Thus, although c-di-GMP and c-di-AMP adopt similar extended-bulge conformations, their binding schemes are very distinct, both in the Hoogsteen-edge H-binding, and in the hydrophobic interactions surrounding the purine bases. Thus mother Nature seems to have evolved protein and RNA in a unique way to bind cyclic di-nucleotides.

3. Page 2, line -3, "obviously many more remain to be identified". Why is it obvious?

AU: The "too many officers, too few soldiers" conundrum is well known to the c-di-GMP researchers. Please see the above-mentioned c-di-GMP census web site and the discussion in recent reviews on c-di-GMP, e.g. by Romling et al.⁷ and by Chou and Galperin⁸.

4. Returning to the riboswitches, the affinity for c-di-GMP of MshEN is not that high. The Class-I c-di-GMP riboswitch binds the second messenger with picomolar Kd. Not that tighter is better, my point being that MshEN binding is not particularly tight.

AU: We agree (please see no. 2 above), but the mode of action of riboswitches is entirely different from that of c-di-GMP-binding protein regulators, so the difference in affinities is hardly surprising. Furthermore, the Kds of this novel c-di-GMP binding domain is at least ten times stronger than that of the PilZ-domain containing proteins. That is something that appears more interesting in the current manuscript.

5. Fig 1F, why is the stoichiometry of the titration not 1:1? Equivalence seems to be attained at about 1:1.5

AU: The stoichiometry of the ITC curve in Fig. 1f is indeed 0.99 as calculated by the commercial software.

6. It would be valuable for the reader if an estimate of the mean coordinate precision (e.g., Luzzati, sigmaA, maximum likelihood) were included somewhere (either Table I or methods).

AU: The maximum likelihood coefficient is 0.19 based on sigma-A value of 0.93 (0.88).

Referee #2 (Remarks to the Author):

A. The key result is that a different motif for binding the bacterial second messenger cyclic di GMP has been found and validated for binding and also not for binding c di AMP. The RLG motif will be an interesting addition informatically to microbiologists who may wish to know the c di GMP binding potential of a protein they work on that lacks a traditional FleQ or PilZ binding site for this messenger. Also there are more circumstantial, but not mechanistically proven, links between pilus fibre expression on the bacterial cell surface and c-di-GMP receptor occupancy.

B. This manuscript doesn't show a mechanism. Therefore it is not a Nature paper. The first FleQ paper discovering that novel c di GMP receptor without full structural data, but with a lot of biology and mechanism was in Mol Micro. The structural biology for FleQ in January this year- was in PNAS. In comparison the biology data in this paper are small and there is no mechanism, although the binding and structure is nice.

In addition the Roelofs paper in PloSPath in 2015 (author's ref # 3) did point to this protein family binding c-di-GMP so this paper comes after that work. This is a nice piece of work and should be published in a good journal but is not as ground breaking in terms of mechanistic understanding as a Nature paper usually is.

C. The pilus assay work doesn't microscopically show the bacterium tested bearing pili (nor actually does the whole paper mention *V. cholerae*- the organism of test). There is no accompanying transcriptional work to test for pilus gene expression, to define the level of regulation effects in the strains showing no pili.

AU: The organism (*V. cholerae*) has been appended in the abstract. The presence of MshA pilus has been reported previously⁹. Using EM and surface pilin ELISA it was demonstrated that pilin secretion requires the presence of MshE. Furthermore, in the same work it was shown that the absence of MshE does not affect MshA abundance but only its secretion. It is possible that absence of MshA in the cell surface or the absence of MshE itself impacts transcription of other pilus genes, however, based on the predicted function of MshE it is more likely that the absence of MshA in the surface is due to impaired ATPase activity.

There is no explanation of the biofilm formation data and at what level biofilm phenotype is affected by c di GMP binding changes. There is no discussion of the apparent additive effect of the L10/58A mutations versus the L10A or L58A alone on biofilm formation. The single L10 mutation and the double mutation affects pilus formation strongly negatively and the L58A alone does not, but the double mutant strongly negatively affects pilus formation.

AU: The conditions for biofilm formation are described in more detail in material and methods. The paragraph describing the physiological effect of point mutations in VcMshEN has been modified to better highlight (p8-p9) and discuss our observations. The differences in the complementation ability of the L10A and the L58A variants are in agreement with their different capacity to bind c-di-GMP *in vitro*.

The L10A single mutation affects biofilm formation a little and the L58A mutation does not, but then additively in the double mutant there is no biofilm formation? How are these different sets of findings, all presented together in Fig 4 for pilus and biofilm to be reconciled mechanistically??

AU: The L10A single mutant has a strong biofilm defect, not as profound as the defect of a G11I or the double L10A-L58A mutants. The mutation in L58A showed no effect in biofilms formed after 24 hours under the conditions tested. The L10A mutant has a 36 fold increase in Kd for c-di-GMP while the L58A mutant has a 10 fold increase. Based on the *in vitro* and *in vivo* results, it can be suggested that the affinity of the L58A mutant for c-di-GMP is still high enough to support biofilm formation. The ITC experiments also clearly showed that a double mutant in L10 and L58 cannot bind c-di-GMP, hence it was expected to be less active than the individual mutants.

Did the mutants and the wild type bacterium grow at similar growth rates and was this determined before the biofilm experiments- not mentioned in the paper.

AU: All strains used grow at similar growth rates. The growth curves of all these strains are shown in Supplementary Fig. 7.

There was (documented) use of DALI (or similar) to identify structural neighbours -what is RMSD to XpsE structure? More context needed on XpsE, noted in body of text that it is non-binder of Cdg.

AU: Superposition of the *VcMshEN* and *XcXpsEN* structures revealed certain differences (with a RMSD of 1.35 Å over 56 residues, p4). The structural differences between *VcMshE* and *XcXpsEN* is also described in p12. The non-binding nature of the *VcMshE* analogue *XcXpsEN* is further described in p10: “However, it is important to note that XpsE (or GspE) from *Xcc17*, an analogue of *VcMshEN*, has lost the ability to bind c-di-GMP due to the changes in several key conserved residues (Fig. 2), such as G11V, I19R, L29R, R38G, G40L, L54H, L58C and Q61V (Fig. 2a).

Figure 3 structural figure is too busy for effective interpretation by non-specialist reader.

AU: I would combine this comment with the subsequent one, “No need for stereo figures in the modern era, I would remove these from main text figures”. Personally I would prefer to presenting structures in stereo in order to get a clearer viewing. For example, although it seems to be busy in Fig. 3, but when viewed in stereo, every feature becomes quite clear. If the editor deems this unnecessary, we can remove the stereo from the figures.

- what was effect on cdg binding of mutation of D108 (completes binding pocket by reaching in from other domain of clamshell fold)?

AU: The D108 residue is conserved among MshE orthologs but not necessarily conserved in other MshEN domain-containing c-di-GMP-binding proteins shown on Fig. 2A, such as PA3740 (which has Leu in the corresponding position) and Dgeo_1755 (ABF46050, which has Ser in that position). Furthermore, in supplementary Fig. 4 we show that the stand-alone MshEN_N subdomain (MshE₁₋₆₂), which does not have that D108 residue, binds c-di-GMP with K_D 0.35 μM, i.e. as good as the full-length MshEN domain.

-no information at all on crystallogensis (e.g. growth condition), hampers reproduction of study but more importantly, removed context from structure (e.g. pH).

AU: A paragraph with a sub-title of “Crystallization of the native and Se-labeled MshEN/c-di-GMP complex” has been appended in p18.

-Which software was used to locate the Selenium sites - no information given.

AU: AutoSol subroutine in the PHENIX program was used to locate three Se-atom positions as described in the subtitle of “*X-ray data collection and structure refinement*” in page 19.

-p.4 (and other) be careful when documenting nature of binding sequence - sometimes "motif" is used for full sequence, other times refers to "half" of the full sequence (i.e. motif itself comprised of two motifs but usage should not be interchangeable for clarity)

AU: Indeed, this novel c-di-GMP binding motif is difficult to describe in a precise way. It contains two 25-residue motifs that each binds half of the c-di-GMP molecule. Two such motifs linked by five residues constitute a complete 53-residue motif for binding the whole c-di-GMP molecule. We will use the 25-residues motif to describe c-di-GMP binding scheme, and use the 53-residue motif to describe the whole MshEN_N domain.

-Has DSF been used to quantify stability of L to A (and other) mutants? If not, it should be to provide information on stability of helical core in context of interpreting ITC results of cdg binding.

AU: It is unclear to authors what this sentence means. Usually peoples only present their ITC data without measuring their DSF changes. When one alters certain residues in a protein, it may destabilize the protein in a big or small way. But in either case, the stabilities of the protein variants will be reflected by their different binding affinities with their ligands. If editor thinks it is necessary, we will supply them later.

-Please report CC half in data collection table - from the I/sigI of your data it would appear that you have truncated the data and are using less than the "full" resolution range?

AU: The CC half values are 99.2 (81.6) and have been appended in p19. We did truncate some poor data with I/sigI less than 2.

-reference required for protocols on p18

AU: We have included references for the ITC and DSF methodologies in page 18.

Minor Points

-mention organism it comes from in abstract!

AU: *Vibrio cholerae* has been appended into the abstract!

-ambiguity p.3 "exhibits significant steric hinderence against" - this doesn't mean anything. Reword.

AU: This phrase has been changed to "causes significant steric clash against"

-repeat on p.3 regarding 4F3H relationship.

AU: c-di-GMP in the VcMshEN complex adopts a bulged conformation more akin to the one in the XcFimX^{EAL}-PilZ complex (4F3H, drawn in yellow carbons in Fig. 1d).

-explain arrows in figure 2 legend.

AU: These arrows have been explained in the Fig. 2 legend: "The three Leu residues forming the hydrophobic cluster of motif I are connected by red double-edge arrows, those in the motif II by blue double-edge arrows."

-The supplementary would benefit from a figure documenting the structural equivalence (or not) of the two half motifs that comprise the full motif. What is in agreement and what is the RMSD?

AU: Please see Supplementary Fig. 5. Amazingly, although the two guanine bases are located in a different environment (Fig. 2a and 2c), the C α ' atoms of the conserved binding residues are aligned very well, with a RMSD of only 0.8 Å over all 25 residues.

-Was the PDBemotif search limited to Leu or did you try a variety of hydrophobic residues with equivalent groups (Ile, Val)?

AU: We specifically searched for Leu residues.

-p.5 should "of the MshEN domain can even reach" read "of the MshEN_N domain can even reach"?

AU: MshEN domain been changed to MshEN_N domain.

D. Perhaps add the Wilson B-factor to data collection table such that refined B-factors can be compared to this.

-reduce the decimal places used for unit cell axes - one is sufficient!

AU: The Wilson B factor value is 16.5 Å² and has been appended in p19.

E. Little is concluded about the function in terms of mechanisms of pilus regulation and link to biofilm effects (if relevant). More about the structure of the fibre-assembling pilus base would need to be included along with a consideration of the interaction of the receptor with the pilus machinery (or other mechanism).

AU: The role of MshE as the extension ATPase of the Type IV MshA pilus has been addressed in our previous report^{9,10}. In the same work it was shown that c-di-GMP levels control pilin production. The goal of the present work is to characterize at the molecular level the novel c-di-GMP binding motif present in MshE as well as to determine if the conserved amino acid residues in this binding motif affect the physiological role of MshE. Nonetheless, as mentioned by the reviewer it will be of great importance to determine how MshE in response to c-di-GMP affects the structure and the assembly of the pilus fiber. The characterization of this mechanism of action will be analyzed in more depth in a future study.

F. Growth rate determination for mutants to validate biofilm data. Determining the mechanism of pilus regulation observed and linking it to your receptor occupancy for a paper in Nature.

AU: All strains used grow at similar growth rates. The growth curves are all these strains are shown in Supplementary Fig. 7.

G. Referencing is appropriate

AU: Thanks!

H. The manuscript is repetitive of the structural information and light on mechanism and insights into the biological function. It doesn't tell a full scientific story, it reports structure and binding and then adds a few biological assays at the end.

AU: The main focus of the manuscript is to describe a very unusual c-di-GMP binding motif that may become crucial for a better understanding of the abundant effects played by this fantastic molecule. After determining its tertiary structure, we then used a series of bioinformatics methods to reveal its wide-spread presence in the bacterial genome data base. Its strong binding with the c-di-GMP (approximately ten time larger than those of the PilZ-containing proteins!), its presence in many different proteins, its presence in bacteria that don't contain known c-di-GMP domain to date, its unique tri-Leu hydrophobic interaction mode etc. indicate the importance of this novel c-di-GMP binding domain.

Referee #3 (Remarks to the Author):

Cyclic di-GMP is a ubiquitous second messenger in Bacteria that has recently started to be investigated intensively. Encoded on the genomes there is usually more than one pair of cyclic di-GMP turnover proteins. A number of cyclic di-GMP protein and RNA receptors has been identified, however, the number of cyclic di-GMP turnover proteins still exceeds the number of cyclic di-GMP receptors suggesting that additional receptors are to be discovered. Among the cyclic di-GMP receptors recently discovered, several have in common an ATPase domain belonging to the P-loop NTPase superfamily. The transcriptional regulator FleQ of *Pseudomonas aeruginosa* with an AAA+ ATPase domain, which binds cyclic di-GMP with amino acids outside the Walker A/B motifs, was the first protein with an ATPase domain to be identified to bind cyclic di-GMP.

In this manuscript the authors surprisingly identified MshEN, in particular MshEN_N located N-terminal of the ATPase domain of MshE, an ATPase of *Vibrio cholerae* associated with type IV pilus formation, as a novel cyclic di-GMP binding domain. The crystal structure of the MshEN domain in complex with cyclic di-GMP allowed then to determine a novel complex 24 residue long cyclic di-GMP binding motif, which can bind cyclic di-GMP with affinities up to as high as those of RNA aptamers. A particular novel feature of the binding motif is the interaction of the second guanine base of cyclic di-GMP with leucines. Conserved residues critical for cyclic di-GMP binding were identified using isothermal calorimetry, but not interaction with cyclic di-AMP was observed. Assessment of biological significance of mutant proteins established a correlation between cyclic di-GMP binding and inhibition of motility, surface pili production and biofilm formation with the exception of a constitutively active mutant.

Bioinformatic analysis revealed that the MshEN domain is highly abundant in bacterial genomes associated also with other domains and is in fact the second most common domain binding cyclic di-GMP. Selected proteins identified binding of cyclic di-GMP to MshEN/MshEN_N domains as a general characteristic of MshEN and allowed the prediction of cyclic di-GMP binding and non-binding MshEN domains.

In general, the manuscript addresses a highly up-to-date topic in microbiology with novel unexpected data. The work is carefully carried out and the manuscript is well written. To highlight the novel findings in this work I suggest to describe and discuss features of the alternative ATPases, which bind cyclic di-GMP.

AU: We agree with this comment and have added sentences highlighting the differences between two c-di-GMP-binding ATPases (p16). Basically, the transcriptional regulator FleQ, whose structure has been recently published in PNAS¹¹, is totally different from MshE in several respects. First, in FleQ, c-di-GMP is being bound by the ATPase domain itself, whereas in MshE the ATPase domain is not involved in binding, as was clearly demonstrated in our previous paper^{9,10}. Second, the c-di-GMP binding site in FleQ is located on its surface, whereas MshEN forms a hydrophobic pocket that uniquely binds c-di-GMP with an almost 10-fold higher affinity than FleQ. Third the FleQ still use the charged amino acid residues such as Arg, Lys, Glu to bind with an intercalated c-di-GMP dimer in a regular binding mode⁸. In our case, the guanine bases are bound with a very non-canonical binding scheme.

In this context I am wondering whether the authors have excluded binding of cyclic di-GMP to the ATPase domain?

AU: Yes, this has been done in our previous paper¹⁰, which showed the existence of independent c-di-GMP-binding and ATP-binding sites on the MshE (Fig. 2), and located the c-di-GMP binding site to the N-terminus of the protein, away from its ATPase domain (Fig. 4).

Also since, at least in the case of type IV pili, at least two distinct ATPases are involved in pilus assembly and retraction, can the authors discuss cyclic di-GMP binding in this physiological context? It would be highly interesting to investigate the effect of cyclic di-GMP binding on type IV pili biosynthesis versus pilin production

AU: This is an important question, but we have chosen to omit this discussion to save space and also because we actually had no new results on this matter beyond what was presented in our previous two papers in PLoS Pathogens^{9,10}. As a matter of fact, the ATPases involved in pilus assembly often contain additional N-terminal domains that are absent in pilus retraction enzymes, which consist solely of the ATPase domain. This issue is discussed in some detail in the review by Korotkov *et al.* in 2012 Nature Reviews Microbiol¹² and other papers from Wim Hol's group. We can only speculate that these N-terminal domains (such as MshEN_N) might be important for pilus biogenesis and/or are involved in some kind of regulation

Specific comments:

p.2, last line: PA14_2940?

AU: Thanks for pointing out this typo. It has been changed to P14_29490.

p.3, second last line: is this the extended conformation?

AU: Yes, it is a new extended conformation (bulge type).

p.4, l.1: ExLxR is not the entire motif(s) required for cyclic di-GMP binding

AU: We are aware that some extra residues such as Glu or Asp residues may incorporate in binding c-di-GMP, but ExLxR motif is so far the most common term used.

p.5, 1.7: I assume, this is not the same MshEN domain, please clarify.

AU: It has been changed to MshEN_N domain.

p.5, 1.8: It would be physiologically relevant to test cyclic GAMP as this cyclic di-nucleotide is present in *V. cholerae*.

AU: This is an important issue that has also been raised by another reviewer. Please see our response to the first reviewer.

p.6, 1.4: It would be relevant to test the ATPase activity of the mutants especially in the case of the constitutively active variant

AU: It is a good point. We are checking their ATPase activities now. Data will be provided later.

p.6, 1.7: Which type of motility?

AU: For the purpose of this study we focused on motility in soft agar plates, we modified the text to make this clear. It has been previously reported that the absence of *mshE* affects near surface motility too.

p.19, 1.15: ...with some amount?

AU: The amount of components used in the DSF experiment were described in more detail in p20 as following: The DSF experiment was carried out on a BioRad qPCR instrument in a buffer comprising 5 μ l of 25 mM Tris pH 7.5, 100 mM NaCl, and 0.2 mM MgCl₂, 18 μ l of 1:1000 dilution of SYPRO Orange Dye, and 2 μ l of 10 μ M proteins in the presence or absence of 1 mM nucleotides in each well.

p.21, 1.19: Quantitative analysis by Comstat is not reported in this work.

AU: Quantitative analysis of biofilm formation by the Comstat software has been appended in the new supplementary Table 9 in the manuscript.

Figure 1 and 3: Arrange a) b) c) d)

AU: Fig. 1 and Fig. 3 have been rearranged as suggested.

Figure 4: delta mshE pMshE or pMshEL10A

AU: We appreciate the reviewer's observation and in response we modified the labels for the biofilm images in Fig. 4.

Extended data table 3: Useful to indicate that PA3740 is a NfrB homologue

AU: It is now appended in the bottom of supplementary Table 3!

Extended data Figure 3a: For clarity, describe misannotation of the MshEN_N domain in the main text.

AU: Have been corrected as suggested.

References;

- 1 Davies, B. W., Bogard, R. W., Young, T. S. & Mekalanos, J. J. Coordinated regulation of accessory genetic elements produces cyclic di-nucleotides for *V. cholerae* Virulence. *Cell* **149**, 358-370 (2012).
- 2 Sun, L., Wu, J., Du, F., Chen, X. & Chen, Z. Cyclic GMP-AMP synthase is a cytosolic DNA sensor that activates the type I interferon pathway. *Science* **339**, 786-791 (2013).
- 3 Ablasser, A. *et al.* cGAS produces a 2'-5'-linked cyclic dinucleotide second messenger that activates STING. *Nature in press* (2013).
- 4 Jones, C. & Ferré-D'Amaré, A. Crystal structure of a c-di-AMP riboswitch reveals an internally pseudo-dimeric RNA. *EMBO J.* **33**, 2692-2703 (2014).
- 5 Ren, A. & Patel, D. c-di-AMP binds the ydaO riboswitch in two pseudo-symmetry-related pockets. *Nat Chem Biol.* **10**, 780-786 (2014).
- 6 Gao, A. & Serganov, A. Structural insights into recognition of c-di-AMP by the ydaO riboswitch. *Nat Chem Biol.* **10**, 787-792 (2014).
- 7 Römling, U., Galperin, M. Y. & Gomelsky, M. Cyclic di-GMP: the first 25 years of a universal bacterial second messenger. *Microbiol. Mol. Biol. Rev.* **77**, 1-52 (2013).
- 8 Chou, S.-H. & Galperin, M. Y. Diversity of c-di-GMP-binding proteins and mechanisms. *J. Bacteriol.* **198**, 32-46 (2016).
- 9 Jones, C. J. *et al.* C-di-GMP regulates motile to sessile transition by modulating MshA pili biogenesis and near-surface motility behavior in *Vibrio cholerae*. *PLoS Pathogens* **11**, e1005068 (2015).
- 10 Roelofs, K. G. *et al.* Systematic identification of cyclic-di-GMP binding proteins in *Vibrio cholerae* reveals a novel class of cyclic-di-GMP-binding ATPases associated with type II secretion systems. *PLoS Pathogens*, e1005232 (2015).
- 11 Matsuyama, B. *et al.* Mechanistic insights into c-di-GMP-dependent control of the biofilm regulator FleQ from *Pseudomonas aeruginosa*. *Proc. Natl. Acad. Sci. USA* **113**, E209-E218 (2016).
- 12 Korotkov, K., Sandkvist, M. & Hol, W. The type II secretion system: biogenesis, molecular architecture and mechanism. *Nat Rev Microbiol.* **10**, 336-351. (2012).

Reviewer #1 (Remarks to the Author):

I had previously reviewed this manuscript, and all my concerns have been addressed.

One minor point, in page 4, it should be "four asymmetric units combine together to form a unit cell comprising the $P2_12_12_1$ space group" not "several asymmetric units).

AU: A similar concern has been raised by Reviewer 2. In this revised version, this sentence has been deleted.

Reviewer #2 (Remarks to the Author):

In general, the response to review is measured and sufficient. Most of my direct comments have been addressed, with the following caveats:

I still recommend the removal of stereo and point to its general disuse in recent times.

AU: As suggested, we have removed the stereo from Fig. 3a and Fig. 3c, but we prefer to keep the stereo rendering in Fig. 2c, due to its complex nature.

The comment "But in either case, the stabilities of the protein variants will be reflected by their different binding affinities with their ligands" is only partly true - stability and affinity are not always clearly related/dissectable, and it would benefit the study to measure stability of the mutants more directly.

AU: As suggested, we have also checked the stabilities of native MshEN_N and its representative active site variants by using the DSF method in the absence or presence of c-di-GMP (Supplementary Table 2). The wild type and all tested representative variants had their T_m values in the 46 – 60 ° C range, indicating that these variants are stable in the absence of c-di-GMP. For the G40I and G11L variants, the presence of c-di-GMP did not change their T_m values, consistent with the poor, if any, binding of the ligand (see Supplementary Table 2). For single (L10A and L25A) or multiple (L54A/L58A, or L10A/L54A/L58A) replacements that preserve the hydrophobic character of the active site residues, the T_m values increased by 5° or more in the presence of c-di-GMP, indicating that these variants could still be stabilized by c-di-GMP binding.

The added CC1/2 information is welcomed, but I still think with an $I/\sigma I$ around 2 and $CC1/2 \gg 50$, there is still an underestimate of their effective resolution.

AU: The structure of MshEN/c-di-GMP complex has been solved to a high resolution of 1.37 Å. Thus we think this is good enough to reveal the unique interactions between c-di-GMP and the surrounding active site residues of MshEN_N domain. Given our goal of characterizing the new c-di-GMP receptor and its binding mode, we do not believe that further improvements in resolution would significantly affect the conclusions of this work.

Could the authors note (even if just in correspondence) whether there is any effect in diversifying their PDBemotif search to include V/I/M/F as well as Leu?

AU: Our initial searches with PDBemOTIF actually looked for Leu, Val, Ile and Arg residues using the [LIV][LIV][LIV]R pattern with guanine, adenine and various nucleotides as ligands. The retrieved hits (4qg4 and other SAMHD1 structures, 1p9b and other adenylosuccinate synthetases, 2qrd, 1uky, 2cdn and related kinases, and so on) were opened in PDBsum, the respective 'List of interactions' links were downloaded and analyzed one-by-one. We could not find any case where aliphatic residues formed a hydrophobic surface that

would stack with a nucleobase. In most cases, retrieved interactions of LIV residues involved main chain nitrogen and oxygen atoms. We then polled our colleagues who are maintaining protein structure databases in NCBI and around the world asking them for such examples. All we heard back were suggestions to look at 1iyb, 4z0g, and other structures that involved stacking by aromatic residues. We are therefore quite confident that the described Leu-Leu-Leu arrangement represents a new kind of nucleotide binding mode; that is why we put it in the title of the paper. At the reviewers request, we tried the [LIVMF][LIVMF][LIVMF]R pattern. It returned several new structures, including and fructose-2,6-bisphosphatase 1c80 and xanthine oxidase 3nvw but again, no structures that we have checked exhibited anything similar to the Leu-Leu-Leu structural motif.

The new inclusion of space group information on page 4 is rather non-standard; P212121 is not written in the correct format, and the statement that the space group arises via inclusion of several ASUs is a truism and not required.

AU: This sentence has been deleted.

The introductory sentence on p.16 about the FleQ structure is nonsensical and needs to be edited.

AU: The description of the FleQ structure in page 16 has been simplified to: “The transcriptional regulator FleQ, whose structure has been recently published, is also different from MshE: in FleQ, c-di-GMP is bound by the ATPase domain itself, whereas in MshE, the ATPase domain is not involved in c-di-GMP binding, as was clearly demonstrated before⁵”.

Reviewer #3 (Remarks to the Author):

This is the revised and resubmitted version of a manuscript previously submitted to Nature. I still find the topic exciting and the manuscript improved.

AU: Thank you! We greatly appreciate the encouragement.

As the authors promise in their answer to provide data on ATPase activity of MshE proteins with relevant mutations in the MshEN domain in a revised version, I am looking forward to this complementation.

AU: We have indeed checked the ATPase activities of full-length MshE protein and some of its active site variants (data not shown). Consistent with the previous report¹⁰, the activity of native MshE could only be enhanced somewhat (less than 10%) by the addition of external c-di-GMP. Our ATPase activity assays showed that the ATPase activities of native MshE and its variants are relatively low and are not responsive to the presence of c-di-GMP (data not shown). However, the *in vivo* biofilm formation by *V. cholerae* is highly sensitive to the presence of c-di-GMP as well as to variations in the c-di-GMP binding residue (Fig. 4). This discrepancy suggests that MshE requires some external factors to form an active ATPase complex. In fact, EpsE, another secretion ATPase from *V. cholerae*, has been reported to exhibit strong ATPase activity (stimulated more than 100-fold) only in the presence of an interacting partner, the inner membrane protein EspL, as well as the presence of phospholipids such as cardiolipin¹³. A similar behaviour has been observed for GspE from *V. vulnificus* and XpsE from *X. campestris*. Thus the difference in the MshE functioning *in vitro* and *in vivo* could be due to the inability of the monomeric MshE to form a stable oligomer in solution (as observed by gel filtration chromatography, data not shown). A membrane protein such as EspL, which contains a cytoplasmic domain for binding ATPase and a membrane

helix to attach to the phospholipid bilayer, may be required to form an active secretion ATPase complex for secreting PilA or MshA monomer out of the cell for pilus formation¹². However, such extra factors for forming the active MshE complex are currently unknown, and more efforts would be required to characterize these effectors and relate the effect of the presence of c-di-GMP and variation of c-di-GMP binding residues with the MshE ATPase activity *in vitro*.

In addition, as *Vibrio cholerae* is not known to produce c-di-AMP, but c-GAMP, the authors should, at least confidentially to the reviewers, report the data of c-GAMP binding to the MshEN domain.

AU: We thank the reviewer for pointing out this important issue. We have tested the c-GAMP binding with MshEN by ITC and included these data in the manuscript as Fig. 1h. Indeed, we did detect weak to moderate binding ($K_D = 3.3 \times 10^{-4}$) of cGAMP with MshEN. This binding affinity is approximately 1000-fold less than that with c-di-GMP ($K_D = 5.0 \times 10^{-7}$) but is definitely better than that with c-di-AMP (non-detectable). We conclude that only guanine base can properly fit into the MshEN nucleotide binding pocket. That said, in the conditions when c-di-GMP is absent and c-GAMP is abundant (as described by Davis *et al.*, 2012¹, for *V. cholerae* entering the host intestinal environment), binding of c-GAMP to MshE might be physiologically relevant.

The discussion about the differences in the binding modes of cyclic di-nucleotides of RNA riboswitches versus the MshEN domain is quite long and can be, at least with respect to the details, moved to a Supplementary description.

AU: We agree and have significantly shortened the comparison of the interactions of cyclic di-nucleotides with the RNA and protein receptors (top of page 14). Some paragraphs have been moved to the supplementary material (Supplementary Fig. 6).

Minor comments

p.2, l.4: ATPases (MshE)?

AU: We have changed the phrase to “two ATPases (*Vibrio cholerae* MshE and its homolog)”.

p.3, l.20: 53-residue motif and p.3, l.21: novel c-di-GMP motif. In other places in the manuscript the 53-residue motif is called motif, which I can agree with. The authors use domain and subdomain in another context.

AU: The motif and domain are difficult to differentiate in the present case, as the mentioned 53-residue motif in fact forms a separate domain to bind c-di-GMP. But this 53-residue domain contains many non-conserved interrupting residues, so we now refer to the conserved residues as a motif, and to the whole 53-residue sequence as domain to make it clearer.

p.3, l.17: Why problem?, maybe 'enigma'

AU: As suggested, we have changed the word to “enigma”.

p.14, l.14: ...mother nature... maybe replaced by a less emotional expression

AU: We have changed the word to simply “nature”.

Supplementary Table 9: are these representative data from one experiment? Biological and technical replicates are not clearly reported.

AU: The data presented in Supplementary Table 9 were from three independent biofilms that were imaged in duplicate. This statement has been appended into the Table 9 legend.

There are some typos in the manuscript on several occasions. E.g. Abstract: p.2, l.16: Bioinformatic analyses; p.14, l.1: ...in more detail...; p.16, l.10/11: sentence not complete; Supplementary Figure 6: ...red dotted lines.

AU: The manuscript has been carefully checked to remove typos.

References:

- 1 Davies, B. W., Bogard, R. W., Young, T. S. & Mekalanos, J. J. Coordinated regulation of accessory genetic elements produces cyclic di-nucleotides for *V. cholerae* Virulence. *Cell* **149**, 358-370 (2012).
- 2 Sun, L., Wu, J., Du, F., Chen, X. & Chen, Z. Cyclic GMP-AMP synthase is a cytosolic DNA sensor that activates the type I interferon pathway. *Science* **339**, 786-791 (2013).
- 3 Ablasser, A. *et al.* cGAS produces a 2'-5'-linked cyclic dinucleotide second messenger that activates STING. *Nature* **in press** (2013).
- 4 Jones, C. & Ferré-D'Amaré, A. Crystal structure of a c-di-AMP riboswitch reveals an internally pseudo-dimeric RNA. *EMBO J.* **33**, 2692-2703 (2014).
- 5 Ren, A. & Patel, D. c-di-AMP binds the ydaO riboswitch in two pseudo-symmetry-related pockets. *Nat Chem Biol.* **10**, 780-786 (2014).
- 6 Gao, A. & Serganov, A. Structural insights into recognition of c-di-AMP by the ydaO riboswitch. *Nat Chem Biol.* **10**, 787-792 (2014).
- 7 Römling, U., Galperin, M. Y. & Gomelsky, M. Cyclic di-GMP: the first 25 years of a universal bacterial second messenger. *Microbiol. Mol. Biol. Rev.* **77**, 1-52 (2013).
- 8 Chou, S.-H. & Galperin, M. Y. Diversity of c-di-GMP-binding proteins and mechanisms. *J. Bacteriol.* **198**, 32-46 (2016).
- 9 Jones, C. J. *et al.* C-di-GMP regulates motile to sessile transition by modulating MshA pili biogenesis and near-surface motility behavior in *Vibrio cholerae*. *PLoS Pathogens* **11**, e1005068 (2015).
- 10 Roelofs, K. G. *et al.* Systematic identification of cyclic-di-GMP binding proteins in *Vibrio cholerae* reveals a novel class of cyclic-di-GMP-binding ATPases associated with type II secretion systems. *PLoS Pathogens*, e1005232 (2015).
- 11 Matsuyama, B. *et al.* Mechanistic insights into c-di-GMP-dependent control of the biofilm regulator FleQ from *Pseudomonas aeruginosa*. *Proc. Natl. Acad. Sci. USA* **113**, E209-E218 (2016).
- 12 Korotkov, K., Sandkvist, M. & Hol, W. The type II secretion system: biogenesis, molecular architecture and mechanism. *Nat Rev Microbiol.* **10**, 336-351. (2012).
- 13 Camberg, J. L. *et al.* Synergistic stimulation of EpsE ATP hydrolysis by EpsL and acidic phospholipids. *EMBO J.* **26**, 19-27 (2007).

Reviewer #2 (Remarks to the Author):

All my comments have been satisfactorily addressed. I look forward to seeing the manuscript in press.

AU: Thanks.

Reviewer #3 (Remarks to the Author):

This is the revised version of a previously submitted manuscript. I have no more comments to make.

AU: Thanks.